# Neural Implicit Action Fields: From Discrete Waypoints to Continuous Functions for Vision-Language-Action Models

Haoyun Liu [1 2 3]  Jianzhuang Zhao [4]  Xinyuan Chang [3]  Tianle Shi [5 2]  Chuanzhang Meng [5 2]  Jiayuan Tan [2]
Feng Xiong [3]  Tong Lin [6 3]  Dongjie Huo [7 3]  Mu Xu [3]  SongLin Dong [2 8]  Zhiheng Ma [2 8]  Yihong Gong [6 2]
Sheng Zhong [1]

## Abstract

Despite the rapid progress of vision-language-action (VLA) models, the prevailing practice of predicting action chunks as discrete waypoints remains structurally misaligned with the intrinsic continuity of physical motion. This discretization arises naturally from fixed-rate robot data collection and the token-by-token prediction paradigm of large language models, but ties actions to rigid sampling rates, does not naturally support analytically consistent higher-order derivatives, and introduces quantization artifacts that hinder precise, compliant interaction. We propose *Neural Implicit Action Fields (NIAF)*, which reformulates chunk-level action representation from discrete waypoints to continuous action functions. Using a vision-language model as a hierarchical spectral modulator over a learnable motion prior, NIAF synthesizes continuous-time action manifolds with arbitrary temporal resolution. This formulation enables analytical differentiation, allowing explicit supervision of velocity and regularization of higher-order derivative signals to promote mathematical consistency, physical plausibility, and control smoothness. Our approach achieves strong results on CALVIN and LIBERO across diverse backbones. Real-world experiments further confirm that NIAF supports stable impedance control, bridging policy-side action generation and execution-side smooth control.

---

[1] State Key Laboratory for Novel Software Technology, Nanjing University [2] Faculty of Computility Microelectronics, Shenzhen University of Advanced Technology [3] Amap, Alibaba Group [4] Istituto Italiano di Tecnologia [5] Shenzhen University [6] Xi'an Jiaotong University [7] Beijing University of Chemical Technology [8] Guangdong Provincial Key Laboratory of Computility Microelectronics. Correspondence to: Zhiheng Ma <mazhiheng@suat-sz.edu.cn>.

*Proceedings of the $43^{rd}$ International Conference on Machine Learning*, Seoul, South Korea. PMLR 306, 2026. Copyright 2026 by the author(s).

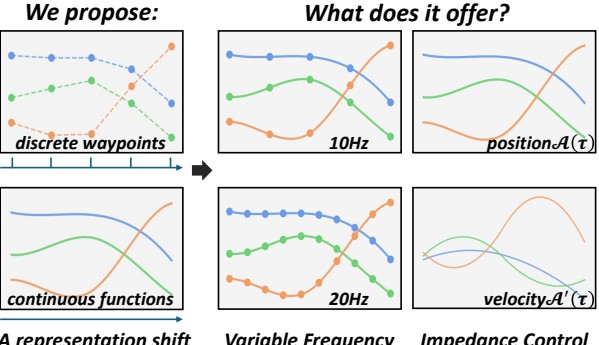

*Figure 1.* **From Discrete Waypoints to Continuous Functions.** Prevalent methods learn temporally discrete waypoints bound to a fixed action sampling rate. We instead model the action trajectory as a continuous time function, which offers two key advantages: 1) Resolution Independence, enabling querying at arbitrary control frequencies without interpolation artifacts; 2) Analytical Differentiability, which allows for explicit velocity supervision and jerk regularization, providing precise and smooth reference profiles required for impedance control.

## 1. Introduction

The convergence of Large Language Models (LLMs) and robotics has catalyzed the emergence of Vision-Language-Action (VLA) models, endowing embodied agents with the capacity to reason about complex semantic instructions and execute diverse manipulation tasks. The paradigm of action modeling in these systems has shifted from predicting instantaneous, single-step tokens to generating fixed-horizon action sequences. Early foundational models, such as RT-2 (Zitkovich et al., 2023) and OpenVLA (Kim et al., 2025b), adopted an autoregressive paradigm, predicting step-wise action tokens. To better capture trajectory coherence and reduce compounding errors, subsequent approaches like ACT (Zhao et al., 2023) and Diffusion Policy (Chi et al., 2025) introduced action chunking, predicting fixed-horizon trajectories as sequences of waypoints. More recently, methods like BEAST (Zhou et al., 2026) and FAST (Pertsch et al., 2025) have further refined this by encoding actions into compressed latent representations like B-spline control points or DCT coefficients to improve compression efficiency.

Despite these advancements, we argue that the prevailing action representation of discrete waypoint prediction remains misaligned with the continuous nature of physical control systems. By reducing smooth physical motions to finite sequences or control points, existing methods encounter three critical limitations:

**1) Rigid Time Discretization.** Current models implicitly bind predictions to a fixed training data frequency. This prevents adaptation to arbitrary control rates and precludes querying trajectories at sub-step resolutions without introducing interpolation artifacts.

**2) Lack of Higher-Order Dynamics Supervision.** While the spline-based tokenizer in BEAST can theoretically ensure continuity, it instead quantizes control points into discrete codebooks and fails to constrain higher-order derivatives. Other methods inherently lack higher-order continuity, yielding discontinuous velocity profiles that can lead to motion jitter and control instability.

**3) Dynamic Inconsistency and Control Incompatibility.** Discrete representations lack analytical differentiability, making it infeasible to simultaneously supervise position and velocity while maintaining mathematical consistency. Moreover, relying on numerical differentiation to recover velocity amplifies quantization noise, rendering the precise feedforward terms required for impedance control unattainable. Consequently, robots are confined to stiff position control, limiting capabilities in speed-sensitive tasks.

To restore the intrinsic continuity of physical motion, we propose **Neural Implicit Action Fields (NIAF)**, shifting from predicting discrete waypoints to continuous action functions. Instead of outputting a finite sequence of coordinates, we model the action chunk trajectory as a parameterized, continuous-time function $\mathcal{A}(\tau) = \Phi(\tau; \theta)$. Leveraging the continuous nature of Implicit Neural Representations (INRs) (Mildenhall et al., 2021), we reformulate action decoding as a parameter prediction problem: the multimodal large language model (MLLM) serves as a hypernetwork that regresses the parameters $\theta$ of an INR, thereby conditioning the action function on the multimodal task context.

To strictly enforce higher-order continuity, we instantiate NIAF using Sinusoidal Representation Networks (SIREN) (Sitzmann et al., 2020), overcoming the derivative limitations of ReLU-based INRs. In this framework, the MLLM acts as a hierarchical Spectral Modulator, offering two key benefits: 1) It guarantees $C^\infty$ smoothness by construction, eliminating quantization artifacts to suppress motion jitter; 2) The learned shared meta-parameters serve as a generic motion prior. By predicting lightweight modulation coefficients instead of full trajectories from scratch, this structured adaptation maximizes learning efficiency and physical plausibility. Our contributions are threefold:

- We propose **Neural Implicit Action Fields**, which reformulates action representation from discrete waypoints prediction to continuous function regression. By positioning the MLLM as a hierarchical spectral modulator over a learnable motion prior, NIAF synthesizes infinite-resolution continuous functions in a single forward pass.

- We introduce an **analytic physics-informed supervision** strategy that exploits the differentiability of our implicit representation to circumvent quantization noise. By explicitly supervising velocity and regularizing jerk, we enforce mathematical consistency between kinematic profiles and provide the analytically precise feedforward signals essential for impedance control.

- We demonstrate NIAF's **scalability and performance** across diverse backbones (Florence-2 to Qwen3-VL), achieving state-of-the-art results on CALVIN and LIBERO benchmarks. Real-world experiments further validate that our continuous representation effectively mitigates control jitter inherent in discrete baselines, enabling superior smoothness in delicate dynamic tasks.

## 2. Related Works

**Action Representations in VLA.** Action modeling has evolved from *step-wise action prediction* in early foundational models such as RT-2 (Zitkovich et al., 2023), RoboFlamingo (Li et al., 2024b) and OpenVLA (Kim et al., 2025b), to *Action Chunking* (e.g., ACT (Zhao et al., 2023) and Diffusion Policy (Chi et al., 2025)) to capture trajectory coherence. To further enhance efficiency, recent parametric methods such as VQ-VLA (Wang et al., 2025b), LipVQ-VAE (Vuong et al., 2025), ActionCodec (Dong et al., 2026), OAT (Liu et al., 2026a), FAST (Pertsch et al., 2025), FASTER (Liu et al., 2026b), OmniSAT (Lyu et al., 2025) and BEAST (Zhou et al., 2026) encode trajectories into compact bases, including VQ codes, ordered discrete tokens, DCT coefficients, or B-spline control points. Beyond trajectory-level compression, another line of work explores *latent action representations*, which infer action-related transition factors from visual state changes and use them as intermediate abstractions for policy learning, as exemplified by UniVLA (Bu et al., 2025), SSM-VLA (Cai et al., 2025), and ALAM (Tang et al., 2026). In parallel, generative approaches such as $\pi_0$ (Black et al., 2024), $\pi_{0.5}$ (Black et al., 2025) and GR00T N1 (Bjorck et al., 2025) use flow-matching-based action generation. However, most action representations remain bound to rigid time discretization at the output level. This reliance on fixed-step values necessitates noisy numerical differentiation for motion regularization, hindering precise control. In contrast, NIAF models actions as implicit, differentiable manifolds, enabling the

direct analytical supervision of higher-order kinematics to ensure mathematical consistency and physical plausibility.

**Implicit Neural Representations (INRs).** INRs parameterize signals as continuous functions, offering infinite resolution and memory efficiency. Beyond their success in NeRF (Mildenhall et al., 2021), INRs have been adapted for robotics to represent trajectories (He et al., 2022) and object geometries (Simeonov et al., 2022). A key enabler is **SIREN** (Sitzmann et al., 2020), which employs periodic sine activations to ensure $C^\infty$ smoothness and well-behaved analytic derivatives. Unlike piecewise-linear ReLU-based baselines, SIREN allows NIAF to synthesize infinite-resolution trajectories and provide analytically precise feed-forward velocity signals. This capability is essential for high-performance impedance control, effectively suppressing the motion jitter and control instability inherent in discrete sampling methods.

**Hypernetworks.** Hypernetworks (Ha et al., 2017) establish a meta-learning paradigm where a primary network generates the weights of a target network. In embodied AI, HyperVLA (Xiong et al., 2025) and HyperTASR (Sun et al., 2025) adopt this to adapt policies or perception modules. Recent advances like Trans-INR (Chen & Wang, 2022) and modulation-based foundation models (Gu & Yeung, 2025) refine this into parameter modulation, where a transformer maps queries to modulation vectors for a shared set of INR meta-parameters. NIAF uniquely instantiates the MLLM as a hierarchical spectral modulator, mapping multimodal context to the frequency and phase spectra of a SIREN. This architecture unifies the high-level semantic reasoning of VLMs with the low-level analytical rigor of INRs, enabling efficient, single-pass synthesis of continuous action functions.

## 3. Method: Neural Implicit Action Fields

In this section, we introduce Neural Implicit Action Fields, a representation framework that models robotic actions as continuous-time functions. As illustrated in Figure 2, NIAF departs from conventional discrete waypoint prediction and instead predicts the parameters of a continuous action function. We first formalize this representation change mathematically.

### 3.1. Problem Formulation: From Discrete Waypoints to Continuous Action Functions

Let $\mathcal{O}$ denote the multimodal observation space (comprising RGB images and proprioception, etc.), and $\mathcal{T}$ denote the natural language instruction space. Our goal is to construct a policy $\pi$ that maps an input $(\mathbf{o}, \mathbf{t}) \in \mathcal{O} \times \mathcal{T}$ to robotic actions.

**Definition 1 (The Discrete Paradigm: Finite Sampling).**

Existing autoregressive or diffusion-based VLA methods typically represent actions in the discrete time domain. Let $T$ denote the physical duration of an action chunk; the model predicts a sequence of discrete points over fixed $H$ time steps:

$$\mathcal{A}_{seq} = \{\mathbf{a}_0, \mathbf{a}_1, \ldots, \mathbf{a}_{H-1}\}. \tag{1}$$

*Limitation:* This representation is tied to a fixed temporal discretization and does not naturally provide consistent derivative information between sampled points without additional interpolation or finite differencing.

**Definition 2 (The Continuous Paradigm: Action-Time Function).** We define the action chunk trajectory as a continuous vector-valued function over a normalized domain $\tau \in [-1, 1]$ corresponding to the chunk duration $T$:

$$\mathcal{A}(\tau) = \Phi(\tau; \boldsymbol{\theta}), \quad \boldsymbol{\theta} \in \Theta. \tag{2}$$

Under this formulation, the objective shifts to predicting the parameters $\boldsymbol{\theta}$ that uniquely define the function. Crucially, this parameterization is resolution-independent, enabling querying at arbitrary temporal resolutions. During inference, we can recover a discrete action sequence $\mathcal{A}$ at any desired resolution by querying the function:

$$\mathbf{a}_k = \Phi(\tau_k; \boldsymbol{\theta}),$$
$$\tau_k = -1 + \frac{2k}{K-1}, \tag{3}$$

where $K$ is the number of query time points and $k \in \{0, \ldots, K-1\}$ is the step index. This continuous formulation fundamentally decouples the action representation from the training discretization, allowing seamless adaptation to diverse control frequencies.

**Definition 3 (NIAF: Parameter Mapping).** Since the function space is infinite-dimensional, we implement Definition 2 via parameter mapping: we learn a mapping function $F_\pi$ that predicts the parameter set $\Theta$ based on the semantic context:

$$\boxed{F_\pi : \mathcal{O} \times \mathcal{T} \to \Theta.} \tag{4}$$

Leveraging the intrinsic differentiability of a sinusoidal representation network, the action chunk trajectory is instantiated as a $C^\infty$ smooth function once $\boldsymbol{\theta}$ is determined. Velocity and higher-order derivatives are thus analytically defined (further detailed in Sec. 3.3). This enables analytical computation of derivative profiles at arbitrary query points, since the predicted velocity is analytically constrained to be the derivative of the predicted position, the two supervision signals regularize each other and help suppress inconsistent noise in the collected trajectories.

### 3.1.1. MULTIMODAL CONTEXT ENCODING

We utilize a pre-trained multimodal large language model as the semantic backbone. To connect context understanding

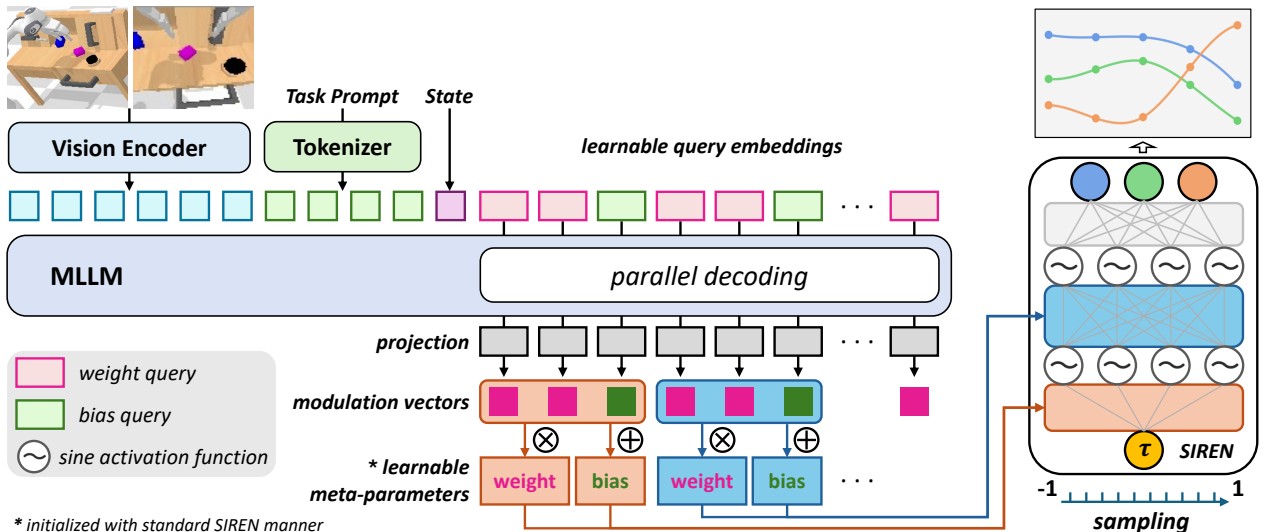

*Figure 2.* **The NIAF architecture.** Instead of predicting discrete waypoints, we reformulate action generation as function regression. Operating as a hypernetwork, the MLLM serves as a hierarchical spectral modulator, transforming learnable query embeddings into modulation vectors conditioned on the multimodal context via one-step parallel decoding. These vectors dynamically reconfigure the shared meta-parameters of a SIREN. Consequently, the instantiated SIREN enables querying actions at arbitrary frequencies by simply sampling the continuous time domain $\tau$.

with continuous action generation, we introduce a set of learnable query embeddings $\mathbf{E}_{qry} \in \mathbb{R}^{Q \times d}$. These query vectors are fed into the MLLM decoder as the final input tokens. By attending to the multimodal context from observations $\mathcal{O}$ and instructions $\mathcal{T}$, the decoder synthesizes the relevant task dynamics into a sequence of modulation latents $\mathbf{Z}$:

$$\mathbf{Z} = \mathrm{MLLM}(\mathbf{E}_{qry}; \mathcal{O}, \mathcal{T}), \qquad (5)$$

where $\mathbf{Z} \in \mathbb{R}^{Q \times d}$. Unlike standard VLAs that predict discrete action bins, we treat $\mathbf{Z}$ as semantic modulation instructions. These tokens do not function as fixed waypoints; they instead provide context-dependent modulation signals that configure the continuous action function.

### 3.1.2. ACTION MANIFOLD DECODING

To instantiate the semantic intent encoded in $\mathbf{Z}$ as a physical trajectory, we employ a sinusoidal representation network as the implicit decoder $\mathcal{A}(\tau) = \Phi(\tau; \boldsymbol{\theta})$.

Crucially, this decoder operates on a hyper-modulation paradigm: instead of learning static weights, the semantic latents $\mathbf{Z}$ are projected into modulation coefficients $(\boldsymbol{\gamma}, \boldsymbol{\beta})$, which dynamically reconfigure the action manifold's geometry. Formally, the forward pass of the decoder generates the action $\mathcal{A}(\tau)$ recursively:

$$\begin{cases} \mathbf{h}^{(0)} = \tau \\ \mathbf{h}^{(\ell)} = \sin\left(\omega_0\left(\hat{\mathbf{W}}^{(\ell)}\mathbf{h}^{(\ell-1)} + \hat{\mathbf{b}}^{(\ell)}\right)\right), \quad \ell = 1, \ldots, L \\ \mathcal{A}(\tau) = \mathbf{W}_{out}\mathbf{h}^{(L)} + \mathbf{b}_{out} \end{cases}$$

$$(6)$$

where $\omega_0$ is a frequency scaling factor. The modulated parameters $(\hat{\mathbf{W}}, \hat{\mathbf{b}})$ are derived from the shared meta-parameters $(\mathbf{W}, \mathbf{b})$ via the instance-specific coefficients:

$$\begin{aligned} \hat{\mathbf{W}}^{(\ell)} &= \mathbf{W}^{(\ell)} \odot (\mathbf{1} + \boldsymbol{\gamma}^{(\ell)}), \\ \hat{\mathbf{b}}^{(\ell)} &= \mathbf{b}^{(\ell)} + \boldsymbol{\beta}^{(\ell)}. \end{aligned} \qquad (7)$$

*Physical Interpretation.* Leveraging the periodic nature of $\phi(x) = \sin(\mathbf{W}x + \mathbf{b})$, $\boldsymbol{\gamma}$ scales the signal's frequency (via $\mathbf{W}$) and $\boldsymbol{\beta}$ shifts its phase (via $\mathbf{b}$). This mechanism effectively decouples the parameter space $\boldsymbol{\theta}$ into a shared meta-prior $(\mathbf{W}, \mathbf{b})$, acting as a stable kinematic backbone, and instance-specific deformations $(\boldsymbol{\gamma}, \boldsymbol{\beta})$, which allow the model to adapt universal motion laws to diverse multimodal contexts.

### 3.1.3. GROUPED HYPER-MODULATION MECHANISM

We now detail the grouped modulation strategy. To circumvent the information bottlenecks inherent in a monolithic global embedding, we distribute the modulation control spatially. By aligning the latent sequence $\mathbf{Z}$ with the SIREN's hierarchical depth, this strategy preserves the granular details of the semantic blueprint essential for precise layer-wise control.

**Structured Token Allocation.** To align the semantic control with the physical network structure, we constrain the latent sequence length to $Q = L \times (G + 1)$. This constraint logically partitions $\mathbf{Z}$ into $L$ sequential blocks, one per SIREN layer. Within the $\ell$-th block, the first $G$ tokens

are routed to control the weights (frequency), while the final token is reserved to control the bias (phase).

**Projection Logic.** The tokens within the $\ell$-th block are projected to modulation coefficients as follows: The $G$ weight tokens are independently projected via an MLP $\psi_\gamma$ and concatenated:

$$\boldsymbol{\gamma}^{(\ell)} = \text{Concat}\left(\psi_{\gamma_1}(\mathbf{Z}_{(\ell,1)}), \ldots, \psi_{\gamma_G}(\mathbf{Z}_{(\ell,G)})\right). \quad (8)$$

The single bias token $\mathbf{Z}_{(\ell,bias)}$ is projected via:

$$\boldsymbol{\beta}^{(\ell)} = \psi_\beta(\mathbf{Z}_{(\ell,bias)}). \quad (9)$$

### 3.2. Analytic Higher-Order Dynamics

A key advantage of NIAF stems from the use of sinusoidal representation networks. Unlike discrete representations that rely on numerical differentiation, SIRENs model the action chunk trajectory as a continuous, infinitely differentiable function. Crucially, the derivative of a SIREN preserves the structure of the original network (since $\frac{d}{dx}\sin(x) = \cos(x)$). This isomorphic derivative structure ensures that dynamic details are preserved analytically, allowing for the computation of higher-order dynamics that are free from the discretization artifacts and quantization errors inherent in finite difference methods.

**Recursive Derivation.** Let the network consist of $L$ layers. We define the pre-activation at layer $l$ as $\mathbf{u}^{(l)} = \hat{\mathbf{W}}^{(l-1)}\mathbf{h}^{(l-1)} + \hat{\mathbf{b}}^{(l-1)}$ and the activation as $\mathbf{h}^{(l)} = \sin(\mathbf{u}^{(l)})$. The analytic velocity $\mathbf{v}(\tau)$ is derived via the chain rule. Starting with the base case $\dot{\mathbf{h}}^{(0)} = 1$, velocity and the underlying layer-wise recursion are formulated as:

$$\begin{aligned} \mathbf{v}(\tau) &= \hat{\mathbf{W}}_{\text{out}}\dot{\mathbf{h}}^{(L)}, \\ \dot{\mathbf{h}}^{(l)} &= \cos(\mathbf{u}^{(l)}) \odot (\hat{\mathbf{W}}^{(l-1)}\dot{\mathbf{h}}^{(l-1)}), \end{aligned} \quad (10)$$

where $\odot$ denotes the Hadamard product. Following this recursive logic, higher-order derivatives such as jerk $\mathbf{j}(\tau)$ are obtained by iteratively differentiating $\dot{\mathbf{h}}^{(l)}$ using the product rule. This formulation promotes mathematical consistency between position and its derivatives, enabling effective physics-informed supervision without the numerical instability of discrete approximations.

### 3.3. Implicit Manifold Regularization and Dynamics Supervision

Our approach unifies kinematic smoothing and dynamics supervision within a single differentiable framework. The continuous parameterization $\mathcal{A}(\tau) = \Phi(\tau; \boldsymbol{\theta})$ naturally ensures $C^\infty$ continuity of the action trajectory. Its intrinsic differentiability allows us to explicitly supervise analytical velocity and regularize jerk, effectively encouraging consistency between the predicted position function and its derivative profiles.

**Implicit Manifold Regularization.** Simulation environments such as CALVIN and LIBERO are characterized by simplified physics and position-only control interfaces, where accurate velocity feedback and feedforward impedance mechanisms are unavailable. We therefore employ a position-only supervision objective:

$$\mathcal{L}_{\text{pos}} = \frac{1}{K}\sum_{k=0}^{K-1}\|\Phi(\tau_k) - \mathbf{a}_{gt,k}\|_2^2, \quad (11)$$

where $\mathbf{a}_{gt,k}$ denotes the ground truth action at time step $k$. By modeling each action chunk as a holistic manifold and regressing the parameters $\boldsymbol{\theta}$ of the SIREN network that represents it, NIAF imposes an intrinsic inductive bias of $C^\infty$ continuity, which strictly enforces temporal coherence. This parameterization also functions as a potent implicit regularizer, inherently suppressing high-frequency quantization artifacts.

**Explicit Dynamics Supervision.** For real-world tasks requiring compliant interaction, supervision on position alone is insufficient. We extend the learning objective to explicitly constrain higher-order motion profiles. Leveraging high-frequency feedback from the robot's FOC drivers, we obtain ground-truth velocity $\mathbf{v}_{gt}$ directly from the internal estimator. Exploiting the analytical differentiability of $\Phi$, we formulate the supervision objective as:

$$\mathcal{L}_{\text{vel}} = \frac{1}{K}\sum_{k=0}^{K-1}\left\|\frac{2}{T}\nabla_\tau\Phi(\tau_k) - \mathbf{v}_{gt,k}\right\|_2^2, \quad (12)$$

$$\mathcal{L}_{\text{jerk}} = \frac{1}{K}\sum_{k=0}^{K-1}\left\|\left(\frac{2}{T}\right)^3\nabla_\tau^3\Phi(\tau_k)\right\|_2^2. \quad (13)$$

where $T$ is the duration of a chunk. The scaling factor $2/T$ arises from the changed time scale $\tau \in [-1, 1]$. Importantly, position and velocity supervision constrain the same SIREN-parameterized action function from complementary measurements. While $\mathcal{L}_{\text{pos}}$ anchors the function value $\Phi(\tau)$, $\mathcal{L}_{\text{vel}}$ anchors its analytical derivative $\frac{2}{T}\nabla_\tau\Phi(\tau)$ using independently measured velocity feedback. This shared functional constraint enforces kinematic consistency between position and velocity, and acts as a cross-signal regularizer that discourages the model from fitting inconsistent high-frequency noise in either measurement.

During inference, the model functions as a high-fidelity reference generator for the impedance control law:

$$\mathbf{u}_{cmd} = \mathbf{K}_p\left(\Phi(\tau) - \mathbf{a}_{curr}\right) + \mathbf{K}_d\left(\frac{2}{T}\nabla_\tau\Phi(\tau) - \mathbf{v}_{curr}\right), \quad (14)$$

where $\mathbf{u}_{cmd}$ denotes the command torque, $\mathbf{a}_{curr}$ and $\mathbf{v}_{curr}$ represent the robot's current position and velocity state derived from proprioception, $\mathbf{K}_p$ and $\mathbf{K}_d$ denote the stiffness

and damping matrices. Specifically, the predicted position $\Phi(\tau)$ sets the time-varying equilibrium point, while the scaled velocity provides a noise-free analytical reference for the damping term. This is a critical advantage over discrete waypoint methods, as explicit velocity supervision allows for stable tracking without stiffening the robot. Furthermore, $\mathcal{L}_{\mathrm{jerk}}$ acts as an auxiliary regularizer to discourage abrupt higher-order variations and promote smoother control transients. The total objective is given by $\mathcal{L}_{\mathrm{real}} = \lambda_p \mathcal{L}_{\mathrm{pos}} + \lambda_v \mathcal{L}_{\mathrm{vel}} + \lambda_j \mathcal{L}_{\mathrm{jerk}}$.

## 4. Experiments

In this section, we evaluate NIAF on two widely used simulation benchmarks and in real-world robotic deployment scenarios. We focus on four aspects: (i) policy improvements from implicit neural representations, (ii) generality of NIAF across different backbone architectures, (iii) its effectiveness on real-world manipulation tasks, and (iv) the benefits of physical-consistency regularization via analytical differentiability.

### 4.1. Experimental Setup and Baselines

**Simulation benchmarks and setup.** We evaluate NIAF on two widely used simulation benchmarks, CALVIN (Mees et al., 2022) and LIBERO (Liu et al., 2023), adopting Florence-2 Large (Xiao et al., 2024) as the backbone following BEAST (Zhou et al., 2026). This model is pretrained exclusively on vision-language data, without prior exposure to large-scale robot datasets, and we perform supervised fine-tuning on the demonstration data collected in simulation. CALVIN comprises 34 tabletop manipulation tasks with a Franka Panda arm across four scene configurations; following common practice, we report results on ABC→D and ABCD→D to measure both in-distribution performance and cross-environment generalization. LIBERO is a lifelong learning benchmark with 130 language-conditioned tasks organized into four suites: Spatial, Object, Goal, and the long-horizon suite Long. Consistent with Flower (Reuss et al., 2025) and BEAST, we train and evaluate separate policies on each LIBERO suite.

**Real-world experiments and setup.** We conduct real-world experiments on two hardware platforms tailored to task complexity: the AgileX Piper 6-DoF arm for single-arm manipulation and the AgileX Cobot Magic for bimanual coordination. Consistent with our simulation setup, we utilize the Florence-2 Large backbone, performing supervised fine-tuning on demonstration data collected via real-robot teleoperation. We evaluate performance across four distinct tasks designed to probe different control capabilities: *Item Placement* and *Cup Stacking* assess basic pick-and-place and contact-rich precision; *Shape Insertion* tests high-precision alignment under tight tolerances; and *Towel*

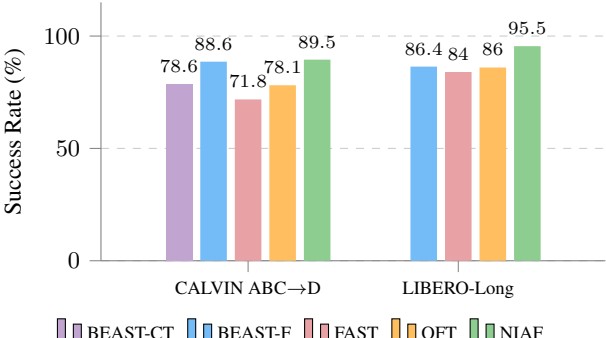

*Figure 3.* **Performance comparison of different action representations** under an identical Florence-2 Large backbone on the most challenging settings.

*Folding* challenges bimanual coordination with deformable objects. Policy inference runs on a remote workstation with an NVIDIA RTX 4090 GPU.

For additional experimental details, please refer to Appendix C.

### 4.2. Simulation Evaluation

**Implicit action representation improves policy quality.** Tables 1 and 2 summarize the results on CALVIN and LIBERO, respectively. NIAF outperforms all baselines, including several large-scale pretrained VLAs with significantly more parameters. On CALVIN, it achieves average chain lengths of **4.47** on ABC→D and **4.66** on ABCD→D. On LIBERO, it attains a 97.9% average success rate, with 100% success on LIBERO-Object and 95.5% on the challenging LIBERO-Long suite. These results validate the superiority of continuous action functions over discrete waypoints. By capturing infinite-resolution trajectory details, our approach achieves state-of-the-art performance and enables robust long-horizon execution with minimal compounding errors.

**Comparison of different action representations.** To fairly evaluate the effectiveness of NIAF, we compare it with BEAST-F, BEAST-CT, OFT (Kim et al., 2025a), and FAST (Pertsch et al., 2025) using an identical Florence-2 Large backbone. These baselines cover a spectrum of distinct action representations. BEAST adopts parallel decoding with learnable queries to predict B-spline control points. Specifically, BEAST-CT maps Florence decoder hidden states to continuous control-point matrices via a linear projection and is trained with an $\ell_1$ regression loss, whereas BEAST-F uniformly quantizes the control-point values into discrete indices in $[0, 255]$ and optimizes a cross-entropy loss. We take the reported BEAST-F and BEAST-CT results from the original paper. OFT follows the same query-based decoding paradigm but predicts action chunks directly via

*Table 1.* **Long-horizon robotic manipulation evaluation on the CALVIN benchmark.** Performance is evaluated under the ABCD→D and ABC→D settings using 1000 evaluation chains. We report the success rate of completing 1–5 tasks consecutively and the average episode length (Avg. Len). Pretraining indicates whether large-scale robot data pretraining is used. Best results are shown in bold.

| Task | Method | Size | Pretraining | Tasks Completed in a Row (1000 chains) | | | | | Avg. Len ↑ |
|---|---|---|---|---|---|---|---|---|---|
| | | | | 1 | 2 | 3 | 4 | 5 | |
| | UP-VLA (Zhang et al., 2025) | 1.3B | ✓ | 0.962 | 0.921 | 0.879 | 0.842 | 0.812 | 4.42 |
| | RoboVLMs (Li et al., 2024a) | 1.6B | ✓ | 0.967 | 0.930 | 0.899 | 0.865 | 0.826 | 4.49 |
| ABCD→D | UniVLA (Wang et al., 2025a) | 9B | ✓ | 0.985 | 0.961 | 0.931 | 0.899 | 0.851 | 4.63 |
| | BEAST (Zhou et al., 2026) | 0.77B | ✗ | 0.981 | 0.962 | 0.930 | 0.893 | 0.848 | 4.61 |
| | FLOWER (Reuss et al., 2025) | 1B | ✗ | 0.989 | 0.967 | 0.939 | 0.902 | 0.855 | 4.62 |
| | **NIAF(ours)** | 0.77B | ✗ | 0.997 | 0.978 | 0.946 | 0.900 | 0.839 | **4.66** |
| | UP-VLA (Zhang et al., 2025) | 1.3B | ✓ | 0.928 | 0.865 | 0.815 | 0.769 | 0.699 | 4.08 |
| | RoboVLMs (Li et al., 2024a) | 1.6B | ✓ | 0.980 | 0.936 | 0.854 | 0.778 | 0.704 | 4.25 |
| ABC→D | UniVLA (Wang et al., 2025a) | 9B | ✓ | 0.989 | 0.948 | 0.890 | 0.828 | 0.751 | 4.41 |
| | BEAST (Zhou et al., 2026) | 0.77B | ✗ | 0.998 | 0.965 | 0.893 | 0.827 | 0.744 | 4.42 |
| | FLOWER (Reuss et al., 2025) | 1B | ✗ | 0.993 | 0.960 | 0.903 | 0.823 | 0.755 | 4.44 |
| | **NIAF(ours)** | 0.77B | ✗ | 0.997 | 0.959 | 0.906 | 0.848 | 0.764 | **4.47** |

*Table 2.* **Experimental Results on the LIBERO Benchmark.** Success rate (%) is reported for each task suite. NIAF is trained and evaluated separately for each suite, best results in each column are shown in bold.

| Method | Size | Pretraining | Spatial | Object | Goal | Long | Average |
|---|---|---|---|---|---|---|---|
| $\pi_0$ (Black et al., 2024) | 3B | ✓ | 96.8 | 98.8 | 95.8 | 85.2 | 94.2 |
| $\pi_0$ FAST (Pertsch et al., 2025) | 3B | ✓ | 96.4 | 96.8 | 88.6 | 60.2 | 85.5 |
| OpenVLA-OFT (Kim et al., 2025a) | 7B | ✓ | 96.2 | 98.2 | 95.6 | 92.0 | 95.5 |
| FLOWER (Reuss et al., 2025) | 1B | ✓ | 97.1 | 96.7 | 95.6 | 93.5 | 95.7 |
| BEAST (Zhou et al., 2026) | 0.77B | ✗ | 92.9 | 97.5 | 93.1 | 86.4 | 92.5 |
| **NIAF (ours)** | 0.77B | ✗ | **98.2** | **100.0** | **98.0** | **95.5** | **97.9** |

*Table 3.* **Comparison of VLA models on LIBERO.** All methods are implemented on the same Qwen3-VL backbone and trained from scratch using a single policy across four task suites. Scores are averaged over 500 trials per suite, all policies use chunking length 8.

| Model | Spatial | Object | Goal | Long | Avg |
|---|---|---|---|---|---|
| Qwen3-VL-FAST | 97.3 | 97.4 | 96.3 | 90.6 | 95.4 |
| Qwen3-VL-OFT | 97.8 | 98.6 | 96.2 | 93.8 | 96.6 |
| Qwen3-VL-PI | 98.8 | **99.6** | 95.8 | 88.4 | 95.7 |
| Qwen3-VL-GR00T | 97.8 | 98.8 | 97.4 | 92.0 | 96.5 |
| **Qwen3-VL-NIAF** | **99.4** | 99.4 | **98.0** | **94.0** | **97.7** |

an MLP projection, while FAST adopts an autoregressive decoding scheme. We report the average success rate on CALVIN ABC→D and the LIBERO-Long suite. As shown in Figure 3, our method consistently outperforms all baselines, highlighting the effectiveness of implicit action representation for robot manipulation. Notably, continuous action space baselines such as BEAST-CT and OFT lag behind NIAF, which rules out the hypothesis that the gains are primarily driven by the absence of discretization and its associated precision loss.

**NIAF on decoder-only MLLM.** To evaluate the scalability of NIAF, we integrate it into the larger-scale, decoder-only Qwen3-VL (Bai et al., 2025). Unlike the iterative denoising required by flow-matching (Bjorck et al., 2025; Black et al., 2024) approaches, NIAF utilizes a query-based design to enable efficient one-step parallel decoding. We train a single multi-task policy across all four LIBERO suites to ensure a fair comparison with baselines from StarVLA (Community, 2026). As shown in Table 3, Qwen3-VL-NIAF achieves a superior average success rate of 97.7%, securing the top performance in three out of four suites.

**Why does implicit representation help in simulation?** Despite the absence of higher-order dynamic feedback in simulation environments, NIAF achieves state-of-the-art performance due to two key factors:

- **Structural Inductive Bias**: By parameterizing action chunks as continuous functions rather than fixed discrete waypoint sequences, NIAF imposes an intrinsic $C^\infty$ continuity bias. This holistic modeling effectively filters out high-frequency quantization noise and ensures temporal coherence even without explicit velocity supervision.

- **Motion-Primitive Priors**: Instead of synthesizing tra-

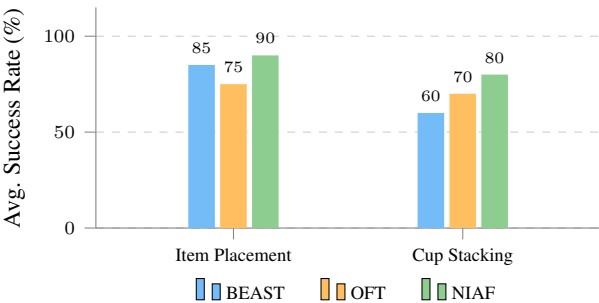

*(a)* Florence-2 backbone: action representation comparison.

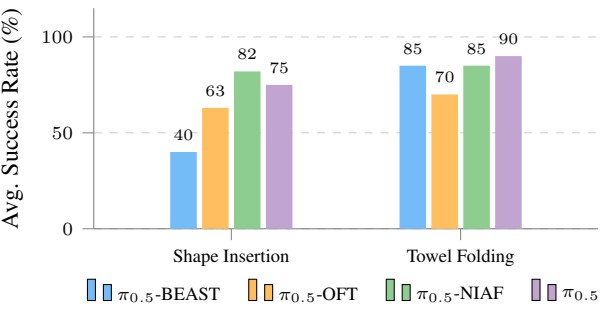

*(b)* $\pi_{0.5}$ backbone: portability evaluation.

*Figure 4.* **Experimental Results on Real-World Robot Tasks.** (a) compares different action representations under the same Florence-2 backbone. (b) evaluates portability by integrating different action heads into the $\pi_{0.5}$ backbone.

jectories from scratch, the model acts as a hierarchical spectral modulator over a learnable motion prior. By predicting lightweight coefficients to deform this prior, it improves learning efficiency and robustness against compounding errors in long-horizon tasks.

## 4.3. Real-World Robot Experiments

We further conduct real-world experiments to validate the effectiveness of NIAF across four distinct manipulation tasks: *Item Placement*, *Cup Stacking*, *Shape Insertion*, and *Towel Folding*. For the first two tasks, we benchmark NIAF against BEAST and OFT using the Florence-2 Large backbone. All methods are trained for 30 epochs per task with identical hyperparameters. To further evaluate portability on the state-of-the-art $\pi_{0.5}$ model ([Black et al.](), [2025]()), we address *Shape Insertion* and the dual-arm *Towel Folding* by integrating different action heads into the $\pi_{0.5}$ backbone. Specifically, $\pi_{0.5}$-NIAF replaces the original iterative flow-matching denoising in the action expert with query-based parallel decoding to generate SIREN parameter-modulation vectors; $\pi_{0.5}$-BEAST replaces the flow-matching target with B-spline control points; and $\pi_{0.5}$-OFT projects each query to one timestep's joint action. To bridge the gap between VLM inference costs and the requirement for high-frequency robot control, we adopt an action chunk length of 50, which poses

challenges for maintaining trajectory fidelity over long horizons. The complete task configurations, data acquisition procedure, and success criteria are provided in Appendix C.

**Real-robot results.** Figure 4 summarizes the evaluation outcomes. NIAF outperforms BEAST and OFT in both tasks, achieving a 90% success rate on *Item Placement* and 80% on *Cup Stacking*. On the $\pi_{0.5}$ backbone, $\pi_{0.5}$-NIAF outperforms all other action heads on the precision-demanding *Shape Insertion* task. We attribute $\pi_{0.5}$-BEAST's poor performance on Shape Insertion to its low-pass filtering nature, which over-smooths the critical micro-adjustments required during insertion. On *Towel Folding*, all action-head replacements underperform vanilla $\pi_{0.5}$, which we attribute to partial forgetting of the pretrained policy's acquired knowledge priors when replacing the action head. We attribute NIAF's overall gains to two advantages of our continuous formulation. First, the SIREN-based action-function parameterization encourages smooth temporal profiles and can reduce high-frequency jitter in the reference trajectory. Second, its analytical differentiability provides velocity references for impedance-style execution in contact-rich tasks such as *Cup Stacking* and *Shape Insertion*.

**Effectiveness of analytic physics-informed supervision.** We visualize the control dynamics of the *Item Placement* task in Figure 5 to evaluate the temporal coherence of the generated policies. BEAST and OFT make impedance control difficult due to noisy finite-difference velocity estimates, forcing the use of stiff position control. Consequently, their monitored velocity profiles exhibit high-frequency jitter that continually fluctuates around zero, indicating that these methods fail to capture the dynamic trends of the demonstrations. This also reflects a limitation of position control that the robot performs disjointed micro-steps at every control cycle.

In contrast, NIAF is trained with simultaneous supervision on both position and velocity. During inference, it generates continuous position and velocity profiles, where the velocity curve maintains consistent non-zero trends. This demonstrates that our explicit velocity supervision successfully encodes physically consistent motion flow rather than merely a sequence of static waypoints. Furthermore, this analytical velocity provides the necessary high-quality feedforward signal that enables the use of stable impedance control, which is otherwise infeasible with noisy discrete baselines. This capability fundamentally validates the advantage of the continuous action function paradigm, proving that our method effectively bridges high-level policy learning and compliant low-level control.

For additional experiments, please refer to Appendix B.

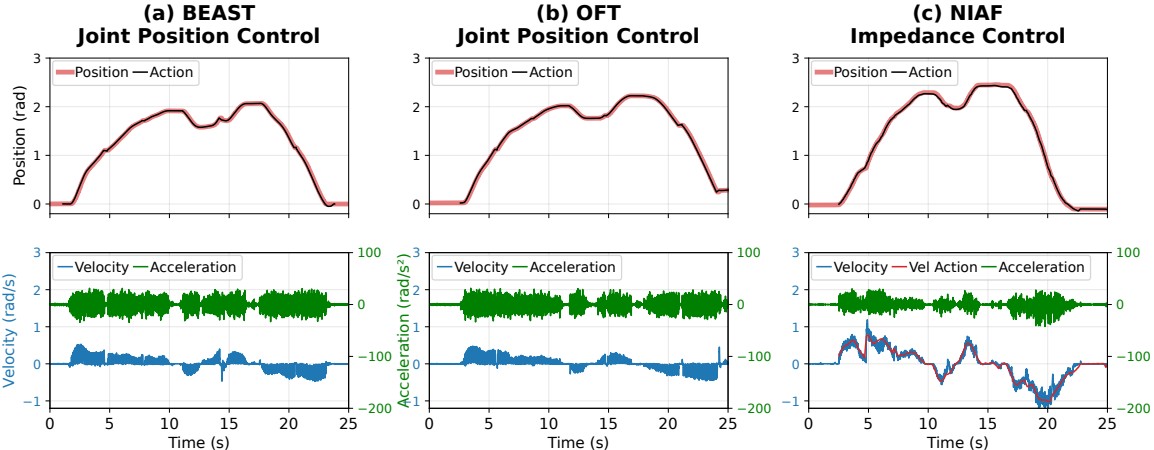

*Figure 5.* **Comparison of control dynamics across different methods.** The top row shows joint position tracking, and the bottom row visualizes velocity and acceleration profiles. **(a) & (b) Baselines (BEAST & OFT):** The velocity profiles exhibit high-frequency oscillations hovering around zero, indicating disjointed stop-and-go motion and poor temporal coherence. **(c) NIAF (Ours, Impedance Control):** In contrast, our method produces a continuous, trend-following velocity reference (red line) that maintains consistent motion without reverting to zero-mean noise. The executed velocity (blue) aligns tightly with this analytical signal, demonstrating effective impedance tracking.

## 5. Conclusion

We introduced Neural Implicit Action Fields (NIAF), a framework that restores the intrinsic continuity of robotic motion by modeling actions as differentiable functions of time rather than discrete waypoints. Our findings suggest that functional parameterization provides a useful structural inductive bias; it effectively regularizes the action space to capture the kinematic coherence that discrete sequences lack. By leveraging spectral modulation over learned motion priors, NIAF enhances learning efficiency and yields the analytically precise signals essential for stable impedance control. Ultimately, this work enables a new practice of action representation capable of fluid, compliant, and resolution-independent physical interaction.

**Limitations.** We identify three main limitations of this work. First, NIAF's primary contribution lies in the action representation and execution side; it does not enhance the high-level reasoning, planning, or zero-shot generalization capabilities of the base VLM. Under matched backbone and training settings, we observe that NIAF is largely neutral with respect to generalization to unseen objects or instructions, which remains driven by the pretrained backbone and data diversity. Second, the explicit velocity supervision in our real-world pipeline relies on high-frequency feedback from the robot's FOC drivers. For low-cost platforms without such sensors, velocity ground truth would need to be estimated from position data via finite differencing, which amplifies teleoperation noise and may degrade the quality of the physics-informed supervision. Third, while NIAF is most beneficial for compliance-sensitive manipulation, long-horizon tasks vulnerable to compounding errors,

and settings requiring flexible execution frequencies, for short-horizon tasks with fixed control rates and no need for smooth velocity references, simpler discrete waypoint representations remain competitive and more straightforward to implement.

## Acknowledgments

This work is supported by the National Natural Science Foundation of China under Grant 62576224, the Shenzhen Key Technical Projects under Grant CJGJZD20220517141605013, JCYJ20220818101406014 and JSGG20220831105801004, the Guangdong Provincial Key Laboratory of Computility Microelectronics 2024B1212010007, the Guangdong Provincial Characteristic Innovation Project of Regular Higher Educational Institutions 2024KTSCX026, and the Research and Development Fund Project of Xi'an Thermal Power Research Institute Co., Ltd. (TP-25-TYK10).

## Impact Statement

This work studies action representation for vision-language-action models in robotic manipulation. Its societal impacts are similar to those of existing robotic methods: improved execution reliability may benefit automation and assistive robotics, while standard concerns regarding workforce displacement and safe deployment in physical environments apply. We emphasize that real-world deployment should follow appropriate safety and human oversight practices.

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

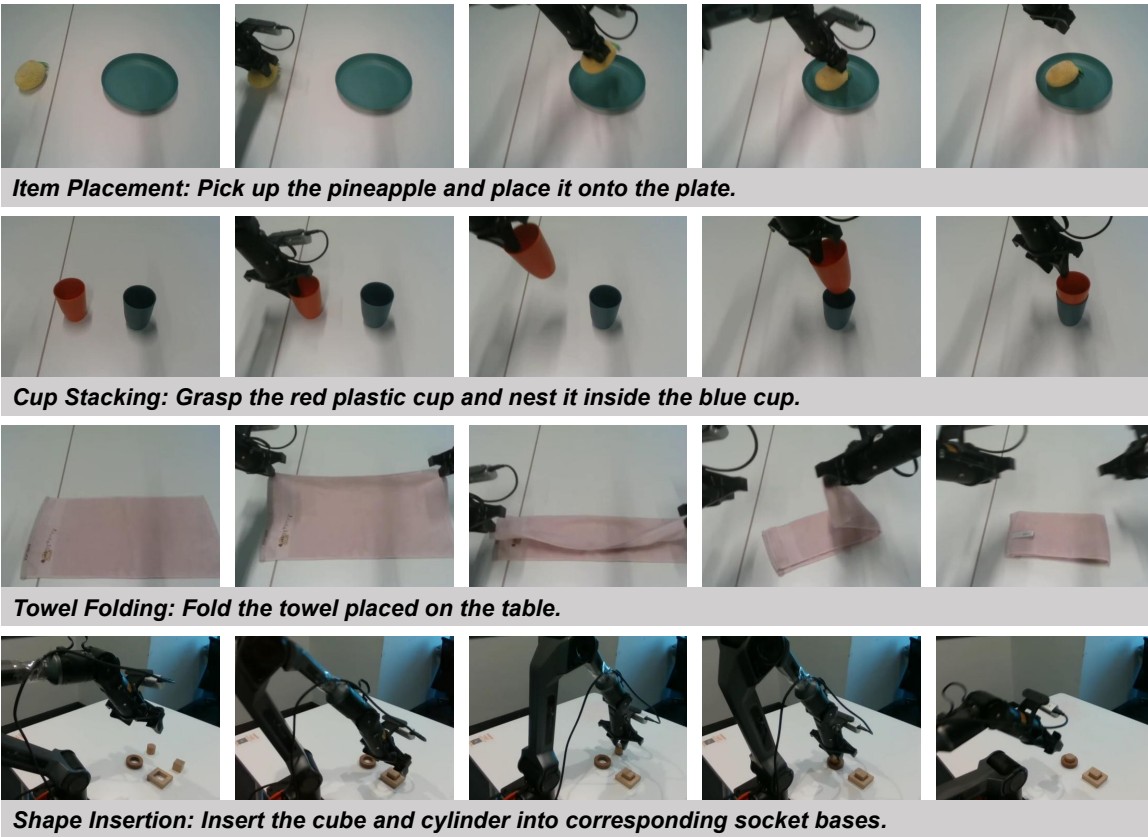

*Figure 6.* **Key frames of real-world evaluation rollouts.** These frames visualize the successful completion of tasks by NIAF during evaluation.

## A. Implementation Details

**Observations:** Inputs consist of synchronized RGB observations, combining a static third-person (simulation) or egocentric (real-robot) view with a wrist view to capture global context and interaction details.

**Action space:** In simulation, supervision targets consist of relative end-effector pose increments paired with binary gripper commands. For real-world tasks, we transition to joint space with continuous gripper control. We employ *chunk-relative trajectories* as supervision targets: each action chunk is constructed using next-step absolute joint configurations and subsequently anchored by subtracting the initial configuration. This delta-from-start scheme effectively mitigates chunk-boundary discontinuities induced by absolute joint-space actions while preserving the physical interpretation of the temporal derivatives of our predicted action function. Leveraging the analytical differentiability of $\mathcal{A}(\tau)$, we incorporate explicit velocity supervision during training (Eq. (12)). Crucially, the control strategies diverge during deployment: while baselines are restricted to joint-position control, NIAF utilizes its analytical structure to simultaneously output joint positions and velocities, thereby enabling impedance control.

## B. Extra Experiments

### B.1. Arbitrary action sampling rate.

As shown in Figure 7, we up-sample the action chunk from 50 to 200 steps and executing at $4\times$ frequency, the velocity profile becomes visibly smoother compared to the lower-frequency baselines. This confirms that our method learns a resolution-independent function, enabling arbitrarily increasing control frequency to mitigate discretization artifacts without retraining, illustrating the advantage of the continuous action-function representation.

*Table 4.* **Hyperparameters for simulation benchmarks.**

| Hyperparameter | LIBERO | | | | CALVIN | |
|---|---|---|---|---|---|---|
| | Spatial | Object | Goal | Long | ABCD→D | ABC→D |
| Action Chunk Length | 10 | 10 | 10 | 10 | 10 | 10 |
| Optimizer | AdamW | AdamW | AdamW | AdamW | AdamW | AdamW |
| Betas | [0.9, 0.95] | [0.9, 0.95] | [0.9, 0.95] | [0.9, 0.95] | [0.9, 0.95] | [0.9, 0.95] |
| Learning Rate | 2e-5 | 2e-5 | 2e-5 | 2e-5 | 1e-5 | 1e-5 |
| Batch Size | 32 | 32 | 32 | 32 | 32 | 32 |
| Number of GPUs | 4 | 4 | 4 | 4 | 2 | 2 |
| Train Steps (k) | 20 | 20 | 20 | 20 | 20 | 20 |

*Table 5.* **Hyperparameters for real-world experiments.**

| Hyperparameter | NIAF
*(Item Placement, Cup Stacking)* | $\pi_{0.5}$-NIAF
*(Towel Folding, Shape Insertion)* |
|---|---|---|
| Action Chunk Length | 50 | 50 |
| Optimizer | AdamW | AdamW |
| Betas | [0.9, 0.95] | [0.9, 0.95] |
| Learning Rate | 1e-5 | 2.5e-5 |
| Batch Size | 16 | 32 |
| Number of GPUs | 4 | 8 |
| Training Duration | 30 Epochs | 30k Steps |

## B.2. Water Cup Transport

To isolate the contribution of explicit dynamic supervision and evaluate NIAF on a smoothness-sensitive task, we conduct a *Water Cup Transport* experiment. The robot must grasp a cup filled with water and transport it to a target location without spilling. This task is highly sensitive to motion smoothness: any abrupt velocity change or jitter during transport causes the water to slosh and spill. We compare four Florence-2-based policies under impedance control: NIAF (pos+vel), trained with both position and velocity supervision; NIAF (pos-only), trained with position supervision only; BEAST (pos-only), the B-spline baseline with position supervision; and FAST (pos-only), the autoregressive baseline with position supervision. For NIAF and BEAST, velocity signals during execution are obtained via analytical derivatives of their respective continuous representations. For FAST, velocity is estimated via finite differencing. The position and velocity profiles are visualized in Figure 8.

Our key observations are: first, regarding representation superiority, NIAF (pos-only) fits fine-grained motions better than BEAST (pos-only), exhibiting fewer high-frequency artifacts in the position profile. Second, NIAF (pos+vel) yields a significantly smoother position curve and exhibits minimal fluctuation in its velocity profile compared to NIAF (pos-only), demonstrating that ground-truth velocity supervision prevents the model from overfitting to the micro-jitters inherent in teleoperation data and forces the policy to learn smoother, more physically compliant motions. Third, adding analytical jerk regularization in NIAF (pos+vel+jerk) further smooths both position and velocity curves, reducing residual high-frequency oscillations without sacrificing positional accuracy. Overall, these results confirm that the SIREN architecture alone provides structural smoothness, but explicit velocity supervision and jerk regularization are necessary to fully exploit the representation's potential for compliant control.

## B.3. Fitting Comparison between SIREN and B-spline

To more precisely characterize the architectural difference between SIREN and B-spline representations, we conduct a controlled fitting experiment. We simultaneously fit position and velocity for a single real-world action chunk of length 200, using a SIREN with 3 hidden layers and width 64 (identical to NIAF's decoder) and a 4th-order B-spline with control points doubled from 10 to 20 to strengthen its fitting capacity. Both are optimized with $\mathcal{L}_{\text{pos}} + \mathcal{L}_{\text{vel}}$ on the same target trajectory.

The fitting results are visualized in Figure 9 and 10. SIREN accurately reconstructs fine-grained local corrections in

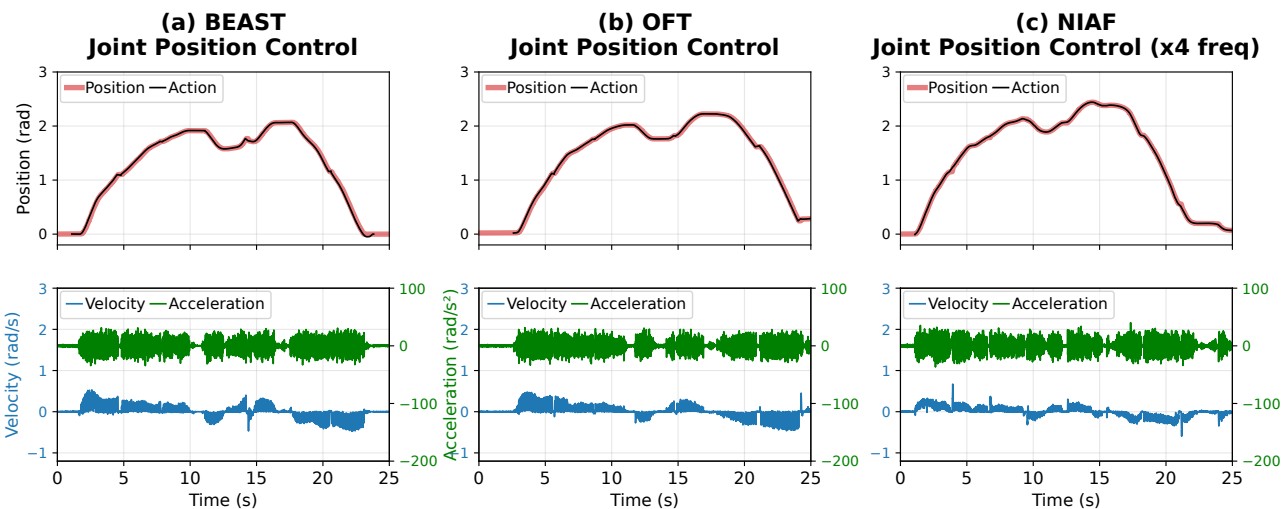

*Figure 7.* **NIAF enables arbitrary action sampling rate.** By upsampling the action chunk from 50 to 200 steps, the velocity fluctuations are reduced comparing to baselines, highlighting the smoothness of the learned manifold.

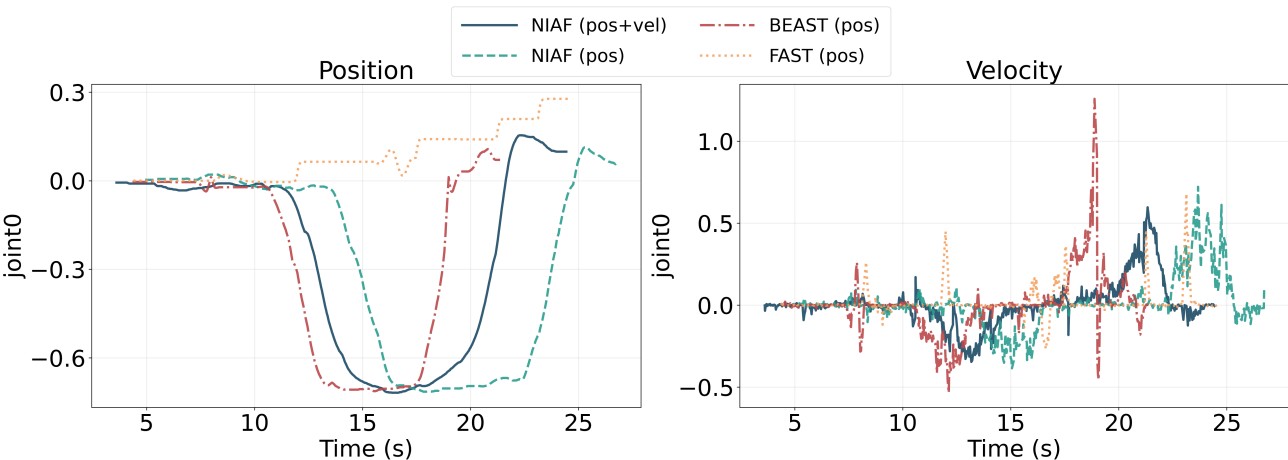

*Figure 8.* **Water Cup Transport: Control dynamics comparison.** Position (top) and velocity (bottom) profiles during transport. NIAF (pos+vel) exhibits the smoothest velocity profile, confirming that explicit velocity supervision effectively suppresses teleoperation jitter.

both position and velocity, and adding jerk regularization further smooths the velocity profile while retaining positional accuracy. In contrast, the B-spline achieves coarse smoothness at the expense of local precision: it underfits critical micro-adjustments even with doubled control-point capacity. This is because guaranteeing $C^2$ or $C^3$ continuity in B-splines requires higher-degree basis functions that widen local support, causing each control point to influence a larger trajectory segment and thereby smoothing away essential fine-grained details. This directly explains $\pi_{0.5}$-BEAST's performance collapse on the precision-demanding Shape Insertion task (Figure 4).

The architectural advantage of NIAF is its ability to **decouple smoothness from expressiveness**: the SIREN's $C^\infty$ continuity is an intrinsic property of the sinusoidal activation, independent of the network's local fitting capacity. This enables high-fidelity local precision while simultaneously allowing analytical smoothness regularization via higher-order derivative terms.

### B.4. Ablation Studies

We conduct ablations on CALVIN ABC→D to analyze key design choices, reporting the average success length in Table 6.

**Action chunk length.** NIAF generates chunks by sampling the continuous function $\mathcal{A}(\tau)$, allowing $H$ to vary without

*Table 6.* **Ablation Studies on CALVIN ABC→D.** We evaluate the impact of key components on the average success length. The default configuration (in **bold**) uses a chunk size $K = 10$, $G = 64$ weight groups, and sine activation functions.

| Ablation Aspect | Variant | Success Length ↑ |
|---|---|---|
| Action Chunk Size ($H$) | 5 | 3.91 |
| | **10** | **4.47** |
| | 15 | 4.22 |
| | 20 | 3.97 |
| Weight Groups ($G$) | 16 | 4.05 |
| | 32 | 4.20 |
| | **64** | **4.47** |
| Activation Function | ReLU | 3.91 |
| | **Sine** | **4.47** |
| SIREN Depth | 2 | 3.24 |
| | **3** | **4.47** |
| | 4 | 4.20 |

altering query dimensionality. As shown in Table 6, performance peaks at $H = 10$, reflecting a trade-off between temporal consistency and motion complexity: excessive $H$ increases the diversity of motion patterns captured within chunks, making optimization more challenging, and increases compounding errors during execution, whereas insufficient $H$ limits the model to short-range corrections, failing to capture coherent long-term trends.

**Weight groups.** Unlike $H$, increasing the weight groups $G$ directly scales the number of decoder queries. We observe a monotonic performance improvement as $G$ increases, confirming that finer-grained modulation of the SIREN meta-parameters enables more precise trajectory reconstruction.

**Activation function.** Replacing SIREN's sinusoidal activation with ReLU leads to a substantial performance degradation. We attribute this to the limited expressivity of ReLU's piecewise linear approximations for smooth motion, while sinusoidal activations provide the spectral fidelity necessary to capture the high-frequency details essential for precise robotic control.

**SIREN depth.** Depth 3 provides sufficient representational capacity while preserving the backbone's limited query token budget; increasing to depth 4 over-parameterizes the SIREN relative to the available modulation signals from the Florence-2 decoder, leading to slight degradation.

## C. Real-World Tasks Setup, Demonstration Collection, and Evaluation

**Common Setup** All real-world experiments are conducted with two synchronized RGB observations: a static front-view camera capturing the global scene and a wrist-mounted camera capturing local interaction. Demonstrations are collected via teleoperation.

**Item Placement** The objective of this task is to pick up a plush pineapple and place it onto a target plate. We collect a dataset of 50 demonstration episodes, where the initial positions of both the pineapple and the plate are randomized to ensure spatial diversity. A trial is considered successful if the robot correctly deposits the pineapple into the plate and the object remains within the plate's area after release. We report the success rate over 20 trials.

**Cup Stacking** In this task, the goal is to grasp a red plastic cup and nest it inside a stationary blue target cup. This operation necessitates precise axial alignment to prevent collision or jamming during the insertion phase. We collect a dataset of 70 demonstrations, randomizing the initial positions of both cups to ensure robustness against spatial variations. Evaluation is performed over 10 trials using a strict binary criterion: a trial is considered successful only if the red cup is fully seated within the blue cup in a stable, upright position, with any toppling or incomplete insertions counted as failures. The overall performance is reported as the success rate.

**Towel Folding** This task evaluates dual-arm manipulation capabilities on deformable objects, demanding flexible bimanual coordination. Starting from a flattened state, the agent must execute two sequential orthogonal folds via distinct collaboration modes: the first fold employs symmetric action where both grippers simultaneously grasp and fold the lateral edges, while

the second requires asymmetric cooperation, where the left gripper anchors the fabric to stabilize it while the right gripper executes the cross-fold. We collect 100 demonstrations and conduct 10 evaluation trials. Performance is reported using a step-wise metric, awarding one point for each successfully completed fold direction. The accumulated score is normalized to a 0–100 scale to represent the average success rate.

**Shape Insertion** The task involves inserting a cube and a cylinder into corresponding socket bases within a trial. Completing tasks requires precise alignment and stable insertion, covering grasping, approach, alignment, insertion, and release. We collect 100 demonstrations. In each trial, the initial object positions are randomized. We evaluate per shape with a 4-level score: 3 points for correct insertion into the matching base, 2 points for incomplete insertion due to inappropriate angles, 1 point for insertion into the wrong base, and 0 points for failed grasps or lifting without placing into any base. The score for each trial is the sum of the scores for both shapes. We report the average score over 10 trials, rescaled to a 0–100 metric as the final success rate.

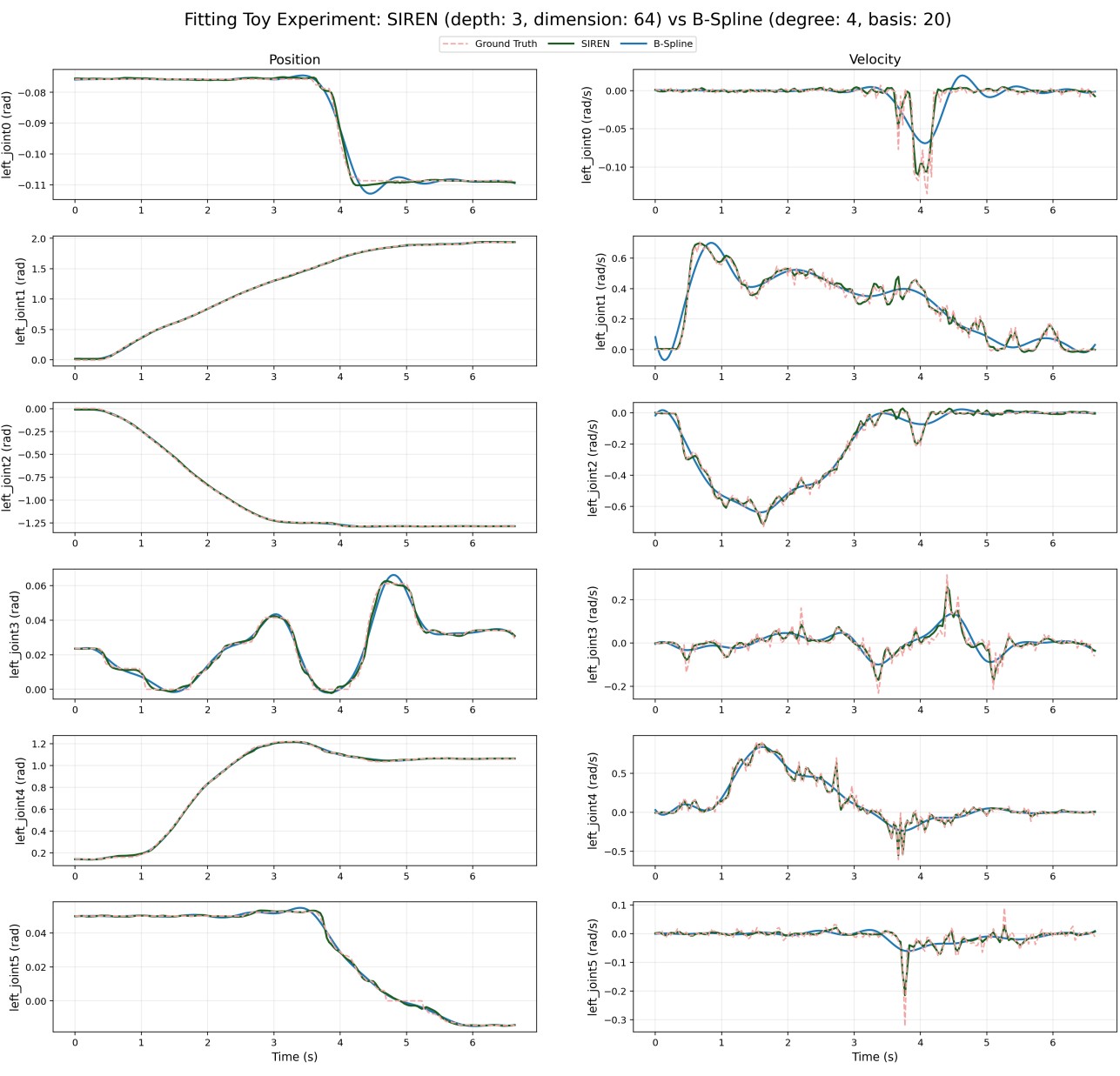

*Figure 9.* **Fitting comparison: SIREN (pos+vel) vs. B-spline (pos+vel).** Both models are fit to the same action chunk with position and velocity supervision. SIREN captures fine-grained local corrections while B-spline acts as a low-pass filter that underfits micro-adjustments.

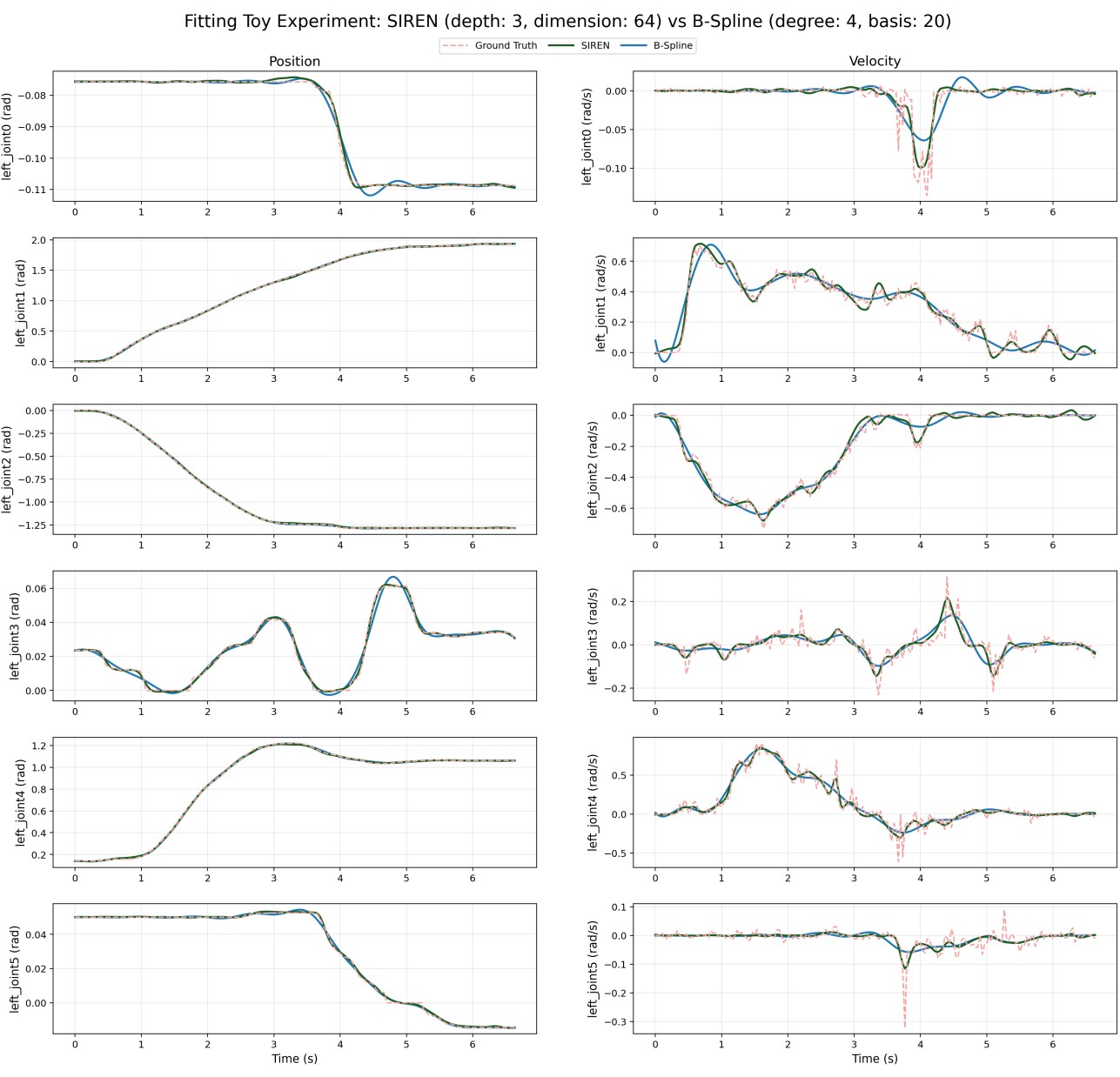

*Figure 10.* **Fitting comparison: SIREN (pos+vel+jerk) vs. B-spline (pos+vel).** Adding analytical jerk regularization further smooths the velocity profile while retaining positional accuracy, demonstrating SIREN's ability to decouple smoothness from expressiveness.

