# OpenReview forum: "Neural Implicit Action Fields: From Discrete Waypoints to Continuous Functions for Vision-Language-Action Models"
_ICML.cc/2026/Conference — ICML 2026 regular_

### Official Review · Reviewer_FYGq · 2026-03-10

**Soundness:** 3
**Presentation:** 3
**Significance:** 3
**Originality:** 3
**Overall Recommendation:** 4
**Confidence:** 4

**Summary:**

In this study, a SIREN model is put forward, where the ReLU activation function employed in end-to-end robot learning models is substituted with sin(·).
The sin(·) features infinite differentiability, which allows it to generate smooth, high-order derivative outputs (including velocity and acceleration) for robot control systems and establish multiple supervision mechanisms.
SIREN can effectively fit continuous physical signals and scenes containing high-frequency details through reasonable weight initialization, and it demonstrates good performance compared with ReLU-MLPs.

**Compliance With Llm Reviewing Policy:**

Affirmed.

**Final Justification:**

The authors’ clarification and additional experiment are helpful. However, given the limitations of acceleration supervision, I will maintain my original score.

**Key Questions For Authors:**

1. The performance on the sim benchmarks looks good, but the real-world tests are relatively toy. Any more real-world results?

2. Most conclusions are based on the simulation results and a limited scale.

3. Discussion on the long-horizon tasks?

**Limitations:**

Yes.

**Strengths And Weaknesses:**

Pros:

+ Good motivation and meaningful contributions.

+ Easy to follow.

+ The method looks sound.

Cons:

- Real-world experiments lack a rigorous ablation study on dynamic loss terms (L_vel, L_acc, L_yerk). It is unclear whether performance gains stem from the SIREN architecture itself or the explicit dynamic supervision. A baseline trained solely with L_pos is necessary to isolate the specific contribution of high-order dynamic losses.

- The framework relies on FOC feedback for velocity vgt. Since acceleration agt is derived via numerical differentiation, the authors should evaluate if vgt can similarly be estimated from position data. This would determine if NIAF's benefits can be extended to low-cost platforms without specialized sensors.

- The paper lacks quantitative analysis on stability when inference sampling significantly exceeds the training action chunk size(H). For a model trained at H=10, the authors should verify if querying at much higher resolutions (e.g., H=50) triggers spectral shifts or high-frequency oscillations in the output.

---

> ### Author Rebuttal · Authors · 2026-03-31
>
> **Reply to Reviewer FYGq**
>
> We sincerely thank the reviewer for the positive assessment and for recognizing the soundness of our method. We address your concerns below.
>
> **Weakness 1 & Q1: Complexity of real-world evaluation and ablation on dynamic supervision.**
>
> We agree that isolating the contribution of explicit dynamic supervision and evaluating NIAF on more complex tasks is critical. To address this, we added a highly smoothness-sensitive Water Cup Transport experiment (please refer to our reply to Reviewer is7m, Q4). While NIAF (pos-only) fits fine-grained motions better than BEAST, it still suffers from noticeable velocity oscillations. NIAF (pos+vel) yields a significantly smoother position curve and exhibits minimal fluctuation in its velocity profile. This demonstrates that ground-truth velocity supervision prevents the model from overfitting to the micro-jitters inherent in teleoperation data (as detailed in our reply to Weakness 2), forcing the policy to learn smoother, more physically compliant motions that accurately reflect the human demonstration.
>
> **Weakness 2: Can $v_{gt}$ used in NIAF be estimated from position data for low-cost platforms without specialized sensors?**
>
> While velocity can mathematically be calculated via finite differences from position data, doing so amplifies inherent teleoperation noise into erratic, unphysical estimates that often exceed physical actuator limits. We therefore explicitly supervise velocity using clean, high-frequency feedback directly from the robot's FOC drivers. Since position and FOC velocity are derived from different hardware estimators but share a strict mathematical derivative relationship, NIAF imposes a physical consistency constraint. This cross-modality regularization forces the model to align its outputs with the true intended motion rather than overfitting to positional noise. Regarding acceleration, we rely on finite differences due to hardware limitations.
>
> **Weakness 3: Stability when inference sampling significantly exceeds training chunk size ($H$)**
>
> We appreciate this question. Querying the action function at a higher resolution does not trigger instability or spectral shifts, as long as the physical duration of the action chunk remains relatively constant. We maintain this duration by proportionally increasing both the sampling and control frequencies. For example, we upsample the action chunk from 50 to 200 points and execute it at a 4x control frequency (Appendix B.1, Figure 7). Since the total execution time for the motion stays the same, denser querying does not "speed up" the robot or introduce high-frequency oscillations. This flexible sampling also potentially enables dynamically adjusting query density near delicate contact phases and unifying heterogeneous pretraining data in physical time. For further discussion on these benefits, please refer to our reply to Reviewer jZrd (Q2).
>
> **Q2: Most conclusions are based on the simulation results and a limited scale.**
>
> We appreciate the reviewer’s perspective on experimental scale. To ensure robustness, we evaluated NIAF across a diverse spectrum of tasks and architectures:
> * **Backbone Scalability:** NIAF shows consistent gains across Florence-2, Qwen3-VL, and $\pi_{0.5}$, suggesting advantages are inherent to the representation.
> * **Simulation for Generality:** Large-scale benchmarks like CALVIN and LIBERO provide the necessary scale for statistically significant and fair comparisons against a wide range of SOTA methods. Our results confirm NIAF's superiority in handling long-horizon tasks and reducing compounding errors.
> * **Real-World for Compliance:** Five real-world tasks, including high-precision Shape Insertion and smoothness-sensitive Water Cup Transport (please refer to our reply to Reviewer is7m, Q4), verify that NIAF’s continuous manifolds effectively bridge the gap between VLA models and physical motions.
> We believe this multi-tiered evaluation provides a comprehensive validation of NIAF's structural superiority and practical scalability.
>
> **Q3: Discussion on the long-horizon tasks.**
>
> We appreciate the opportunity to clarify this point. To be precise, our method does not enhance the high-level reasoning, long-horizon planning, or task decomposition capabilities of the base VLM. The core contribution of NIAF lies in the decoding and execution sides. However, our substantial improvements on long-horizon benchmarks CALVIN ABC$\rightarrow$D and LIBERO-Long demonstrate that long-horizon success is deeply tied to execution quality. NIAF enables highly precise, fine-grained action reconstruction, effectively reducing compounding errors over extended horizons. Please refer to our reply to Reviewer jZrd (Q1) for further discussion on this.
>
> Thank you again for the constructive suggestions.

---

> > ### Author Rebuttal · Reviewer_FYGq · 2026-04-03
> >
> > Thank you for the responses. Since the acceleration target is estimated rather than directly measured, I still think the benefit of using it as a higher-order supervision term should be evaluated more carefully. Overall, I will keep my original score.

---

> > > ### Author Response · Authors · 2026-04-06
> > >
> > > We sincerely thank the reviewer for the continued support and constructive feedback.
> > >
> > > We agree with your insight regarding acceleration supervision. Because it is derived via numerical differentiation rather than directly measured due to our hardware limitations, it inherently contains noise and should be used more carefully. Position and velocity are the primary and most reliable supervision signals, as they have precise sensor ground truth.
> > >
> > > To evaluate the benefit of higher-order terms more carefully, we have conducted a new set of experiments in our follow-up reply to **Reviewer is7m (Q2)**. Specifically, we introduced an analytical jerk regularization term. As visually demonstrated in that thread, this higher-order regularization effectively improves the smoothness of both the position and velocity profiles.
> > >
> > > We will clarify the limitations of acceleration targets and include the new jerk regularization results in the revised manuscript. Thank you again for your time and valuable guidance, which has greatly strengthened our work!

---

### Official Review · Reviewer_dds7 · 2026-03-12

**Soundness:** 3
**Presentation:** 4
**Significance:** 3
**Originality:** 3
**Overall Recommendation:** 4
**Confidence:** 2

**Summary:**

This paper proposes Neural Implicit Action Fields (NIAF), which replaces the standard VLA practice of predicting discrete action waypoints with predicting a continuous-time action function. The core idea is to use a VLM as a hypernetwork (spectral modulator) that outputs modulation coefficients for a shared SIREN implicit neural representation. Then the robot action trajectory can be queried at arbitrary temporal resolution and differentiated analytically to obtain velocity, acceleration, and jerk. The paper argues this gives both a better structural prior for policy learning and cleaner signals for compliant low-level control. Empirically, it reports strong results on CALVIN and LIBERO, shows scalability from Florence-2 Large to Qwen3-VL, and presents real-robot improvements on several tasks, with smoother velocity profiles that support impedance control.

**Compliance With Llm Reviewing Policy:**

Affirmed.

**Final Justification:**

Overall, I find this paper to be clearly written and generally well executed. The method is presented in a coherent way, and the paper includes a reasonable amount of experimental evidence and analysis, especially after the rebuttal.

My main remaining concern is the significance of the contribution. In particular, the added results on unseen object variants and unseen language variants do not show clear improvement in generalization, which I view as an important goal in VLA research. While the authors argue that generalization mainly comes from the underlying VLM, this also makes the added value of NIAF appear more limited. On the positive side, NIAF does not seem to hurt performance, which supports the paper’s soundness.

**Key Questions For Authors:**

- In the real-world experiments, NIAF is fine-tuned for 30 epochs per task. How well does it generalize in zero-shot settings?
- Based on your experiments, can you more clearly characterize when NIAF works best and when discrete waypoint representations may still be preferable?

**Limitations:**

The Impact Statement section is missing.

**Strengths And Weaknesses:**

## Strengths
- The paper identifies a concrete mismatch between discrete waypoint prediction and the continuous nature of physical robot control.
- The paper includes ablations and additional validation that support the claim that the continuous representation contributes to the observed gains.
- The core idea: from "discrete waypoints are mismatched to physical motion" to "predict the action function itself", is clearly written and easy to follow.
- The central idea is genuinely novel at the VLA action-representation level: the model predicts the parameters of a continuous action function, which is a meaningful conceptual shift.
- The paper provides an original control-oriented perspective by using analytical differentiability to supervise velocity, acceleration, and jerk, and to support impedance control.

## Weaknesses
- The empirical evidence is not yet sufficient to show when NIAF is broadly preferable to discrete waypoint methods, and when simpler discrete formulations may remain competitive.
- The paper would be more balanced if it included a clearer discussion of failure cases.
- The novelty lies more in the overall formulation and integration than in the individual components.

---

> ### Author Rebuttal · Authors · 2026-03-31
>
> **Reply to Reviewer dds7**
>
> We sincerely thank the reviewer for the positive and constructive feedback, and for recognizing our reformulation of action prediction as a meaningful conceptual contribution. We address your questions below.
>
> **Q1: In real-world experiments, NIAF is fine-tuned for 30 epochs per task. How well does it generalize in zero-shot settings?**
>
> We appreciate this important question. Based on our current evidence, zero-shot robustness is primarily driven by the VLA backbone's pretraining coverage and model capacity, rather than by the action representation. This is not a limitation specific to NIAF, it reflects the current state of the field.
>
> With the 0.77B Florence-2 backbone that lacks large-scale robot data pretraining, and the limited diversity of our real-world demonstrations comparing to simulation benchmarks, both NIAF and discrete baselines remain sensitive to lighting changes, background clutter, viewpoint shifts, and unseen objects or instructions. When combined with the stronger $\pi 0.5$ backbone, the policy becomes more robust to visual distractors, but we believe this improvement primarily comes from the backbone, not from NIAF.
>
> Our current view is that NIAF is largely orthogonal to zero-shot generalization. Its main contribution is on the decoding and execution side: representing a chunk-level motion as a continuous function, improving temporal consistency, and providing analytically usable derivative signals. We will clarify this in the revised paper.
>
> **Weakness 1 & Q2: When does NIAF work best, and when might discrete waypoint representations be preferable?**
>
> We agree that this boundary should be stated more clearly. Based on our experiments and the structure of the method, NIAF is most beneficial in the following settings:
> * **Compliance sensitive manipulation.** Since NIAF predicts a continuous-time action function, it provides analytically derived velocities consistent with the predicted motion, which is well suited to impedance control (see the newly added Water Cup Transport experiment in our reply to Reviewer is7m, Q4).
> * **Long-horizon tasks vulnerable to compounding errors.** Its benefit here comes from more accurate chunk-level action reconstruction: NIAF uses a continuous function family learned end-to-end rather than a fixed basis, it fits demonstrated trajectories more precisely and reduces local errors during rollout. We believe this is one reason NIAF achieves the strongest results among non-pretrained methods on CALVIN and consistently improves LIBERO-Long with both Florence-2 and Qwen3-VL backbones (see our reply to Reviewer jZrd, Q1).
> * **Settings that benefit from flexible execution frequencies.** The continuous action function allows arbitrary temporal querying during execution, enabling dynamically adjusting the query density near delicate contact phases or under different control rates without changing the predicted motion itself (see our reply to Reviewer jZrd, Q2).
>
> At the same time, for short-horizon tasks with fixed execution rates and no need for smooth velocity references or compliant control, simpler discrete waypoint representations remain competitive and more straightforward to implement. We will state this more explicitly in the revision.
>
> **Weakness 2: Failure cases**
>
> Based on our experiments, NIAF does not fail in a completely new way compared with existing VLAs. Its failures mainly reflect common VLA weaknesses, such as sensitivity to training-data diversity and limited robustness to distribution shifts. As noted in Q1, success rates drop when lighting or viewpoint differs from training. We observe this clearly in Towel Folding: Our initial collected demonstrations mainly cover clean folds, with limited wrinkled or partially folded intermediate states, so the policy may produce unstable actions once execution deviates into these unseen configurations. We view this as a broader challenge in current VLA-based deformable manipulation rather than a failure mode unique to NIAF.
>
> **Impact Statement**
>
> We also appreciate you pointing out the missing Impact Statement, we will include the following Impact Statement:
>
> This paper presents work improves robotic execution reliability benefits automation, it carries standard societal considerations regarding workforce shifts. Our research focuses strictly on low-level action representation and introduces no unique ethical vulnerabilities beyond those inherent to general robotic control.
>
> Thank you again for the constructive feedback. We will revise the paper to better clarify the scope of NIAF, its most effective application regimes, and its representative failure cases.

---

> > ### Author Rebuttal · Reviewer_dds7 · 2026-04-05
> >
> > The rebuttal is helpful and addresses my main concerns. However, it mainly clarifies the scope of the claims rather than adding new evidence. In particular, while it is reasonable to argue that zero-shot robustness is largely determined by the backbone and largely orthogonal to NIAF, explicit zero-shot comparisons under matched backbones and training settings would be needed to show whether NIAF helps, hurts, or is neutral with respect to generalization. Thus, while the response increases my confidence in the work’s potential, I prefer to keep my score.

---

> > > ### Author Response · Authors · 2026-04-08
> > >
> > > Thank you for the helpful follow-up. To better address zero-shot behavior under matched settings, we ran additional zero-shot tests comparing $\pi_{0.5}$ and $\pi_{0.5}$-NIAF on real-world Shape Insertion, using the same backbone, demonstrations, and SFT setup.
> > >
> > > **Unseen object variants.**
> > >
> > > We evaluate two variants, and place one unseen object and its matching base on the table per trial: a pentagonal prism with a matching pentagonal socket, and a Coca-Cola bottle with a hollow tape roll as the target receptacle. Since these unseen variants do not fit the original shape-insertion scoring protocol well, we report three interpretable metrics instead: successful grasp, correct insertion tendency, and successful insertion.
> > >
> > > | Method | Variant | Successful Grasp | Correct Insertion Tendency | Successful Insertion |
> > > | :--- | :--- | :---: | :---: | :---: |
> > > | $\pi_{0.5}$ | Pentagonal prism | 10/10 | 10/10 | 2/10 |
> > > | $\pi_{0.5}$-NIAF | Pentagonal prism | 10/10 | 10/10 | 1/10 |
> > > | $\pi_{0.5}$ | Bottle | 6/10 | 6/10 | 0/10 |
> > > | $\pi_{0.5}$-NIAF | Bottle | 6/10 | 6/10 | 0/10 |
> > >
> > > Both policies still produce grasp attempts toward the correct target, suggesting that after SFT, both policies retain some ability to recognize grasp targets and hole-like receptacles in unseen variants. We attribute this mainly to the pretrained backbone, while NIAF does not appear to significantly change this behavior.
> > >
> > > **Unseen language variants.**
> > >
> > > In the collected demonstrations, the task instruction is “Insert the cube and cylinder into corresponding socket bases. (instruction 0)” We then test two unseen paraphrases: “Place the two blocks into their matching sockets. (instruction 1)” and “Only insert the cube into its matching socket. (instruction 2)”. Each trial in this setting uses the same object configuration as in the demonstrations (a cube and a cylinder).
> > >
> > > | Method | Variant | Successful Grasp | Correct Insertion Tendency | Successful Insertion |
> > > | :--- | :--- | :---: | :---: | :---: |
> > > | $\pi_{0.5}$ | instruction 0 | 18/20 | 15/20 | 12/20 |
> > > | $\pi_{0.5}$-NIAF | instruction 0 | 19/20 | 16/20 | 14/20 |
> > > | $\pi_{0.5}$ | instruction 1 | 17/20 | 13/20 | 9/20 |
> > > | $\pi_{0.5}$-NIAF | instruction 1 | 18/20 | 13/20 | 10/20 |
> > > | $\pi_{0.5}$ | instruction 2 | 17/20 | 13/20 | 9/20 |
> > > | $\pi_{0.5}$-NIAF | instruction 2 | 18/20 | 12/20 | 9/20 |
> > >
> > > Under both unseen instructions, $\pi_{0.5}$ and $\pi_{0.5}$-NIAF still partially succeed at inserting both objects, but performance decreases relative to the seen instruction. The two unseen instructions also lead to similar outcomes, suggesting that neither policy reliably adapts its behavior to the semantic difference between them. This indicates that after task-specific SFT on a fixed demonstration distribution, both policies overfit to the dominant demonstrated routine, tend to ignore fine-grained differences in language instructions, and consequently lose part of their instruction-following ability.
> > >
> > > **Conclusion.**
> > >
> > > These additional experiments more precisely define the role of NIAF with respect to zero-shot generalization: under matched backbone and training settings, NIAF appears largely neutral. Its primary contribution remains on the action representation and execution side, while zero-shot behavior is driven more strongly by the pretrained backbone and the diversity of fine-tuning data.
> > >
> > > We are sincerely grateful for the reviewer’s questions across both rounds, which prompted a more careful clarification of the boundaries of our claims and improved the calibration of the paper. Thank you again for the thoughtful feedback, which has greatly strengthened our work!

---

### Official Review · Reviewer_is7m · 2026-03-12

**Soundness:** 2
**Presentation:** 3
**Significance:** 2
**Originality:** 2
**Overall Recommendation:** 3
**Confidence:** 3

**Summary:**

The paper proposes NIAF, a framework that represents robot action predictions as continuous-time functions instead of discrete waypoint sequences in vision-language-action (VLA) models. Current VLA policies normally predict fixed-length trajectories with discrete actions or control points. However, such discretization introduces issues such as fixed sampling rates and lack of higher-order differentiability, making it difficult to obtain consistent velocity estimates needed for physical control. The proposed method models the action trajectory as a continuous function using an implicit neural representation (specifically SIREN). The MLLM is modified to predict the weights of the INR (through modulation vectors) instead of predicting discrete waypoints. The MLLM thus behaves as a hypernetwork that modulates the weights of a SIREN hyponetwork with a shared learnable motion prior. The approach is evaluated on simulation benchmarks (CALVIN, LIBERO) as well as real-world manipulation tasks, and is compared against trajectory representation approaches such as BEAST, FAST, and OFT.

**Compliance With Llm Reviewing Policy:**

Affirmed.

**Final Justification:**

I appreciate the authors' engagement and efforts to address my concerns during the rebuttal process. As mentioned in my final comment, it is still not conclusively proven whether the higher order derivatives in the proposed method make the position and velocity curves smoother. I am providing a final rating of weak accept to indicate there is room for improvement and clarity. I hope the authors will revise the strong claims made in the paper about the rich benefits of smoothness resulting from implicit incorporation of higher order derivatives.

**Key Questions For Authors:**

1. Why are different baselines used across the real-world tasks in Figure 4? Do OFT and BEAST fail on Shape Insertion and Towel Folding, and if so what are their results?
2. How does the method scale with trajectory horizon or longer action chunks?
3. How sensitive is the approach to SIREN architecture choices (e.g., depth, frequency parameters)?
4. Why are higher-order derivatives (e.g., jerk) necessary for these tasks? B-spline trajectories already provide $C^2$ continuity (smooth position, velocity, and acceleration), which should be sufficient for most robot control pipelines.

**Limitations:**

Yes

**Strengths And Weaknesses:**

Strengths:
- Soundness: Representing trajectories with INRs and predicting their parameters through a hypernetwork is a natural extension of existing INR work. NRs allow querying trajectories at arbitrary temporal resolutions (theoretically infinite frequency). Figure 5 demonstrates the continuity this lends to control dynamics and its use in VLAs.
- Presentation: The paper clearly motivates the limitations of discrete waypoint prediction in VLA models and intuitive argues for modeling actions as continuous trajectories instead of fixed-step sequences
- Significance: The idea addresses a real structural issue in current VLA models: trajectory discretization and lack of smooth dynamics.
- Experiments: Improvements on LIBERO appear relatively strong, particularly on LIBERO Long. Ablations comparing different action representations under the same backbone partially isolate the contribution of the representation.

Weaknesses:
- In Figure 4, OFT, BEAST, and NIAF are compared on Item Placement and Cup Stacking, but Shape Insertion and Towel Folding instead compare $\pi_{0.5}$ and $\pi_{0.5}$ + NIAF. It brings into question why all baselines are not evaluated on all tasks. If OFT and BEAST fail on the harder tasks, this should be reported. The absence of same tasks, same models, & same evaluation protocol settings makes the figure harder to interpret. If as claimed, discrete waypoint policies struggle with compliant control because velocity estimates are noisy, showing BEAST/OFT failing on the harder tasks would actually strengthen the authors’ argument.
- Improvements in Table 1 appear very small (e.g., 4.66 vs 4.62 average chain length in ABCD→D). Considering that in the CALVIN evaluation, each episode attempts 5 tasks sequentially, does a difference of 0.04 tasks corresponds to roughly: 0.04 / 5 = 0.8% improvement? Without significance analysis, it is difficult to assess whether the difference is meaningful.
- The paper’s motivation focuses on improved control dynamics (e.g., impedance control, higher-frequency control), but the evaluation mainly reports task success rates. The real-world tasks shown are relatively simple and do not clearly demonstrate the claimed advantages. It would be more convincing to evaluate scenarios where higher-frequency control or compliance matters. NIAF requires the policy to predict neural network weights rather than trajectories directly, introducing a harder learning problem. The paper does not clearly justify why this added complexity is necessary compared to simpler continuous parameterizations.

---

> ### Author Rebuttal · Authors · 2026-03-31
>
> **Reply to reviewer is7m**
>
> We thank the reviewer for the constructive feedback and for recognizing NIAF's significance in addressing trajectory discretization. Below, we address your comments.
>
> **Weakness1 & Q1: Why are different baselines used in Figure 4? Do OFT and BEAST fail on Shape Insertion and Towel Folding?**
>
> We apologize for the confusing presentation, Figure 4 combines two distinct evaluations, which we will split into two tables for clarity. The first two tasks compare representations (OFT, BEAST, NIAF) using the same Florence-2 backbone. The latter two harder tasks test NIAF's portability by integrating it into the strong $\pi_0$ backbone (replacing flow matching). OFT and BEAST were excluded from these latter tasks as they were outside the scope of this portability test.
>
> **Q2: How does the method scale with trajectory horizon or longer action chunks?**
>
> Our ablation study (Appendix B.2, Table 6) reveals a trade-off in chunk size: short chunks ($H=5$) struggle to capture coherent long-term trends, while overly long chunks ($H=20$) complicate optimization and amplify compounding errors. Thus, we adapt $H$ related to the environment's control frequency (10 in simulation, 50 in real-world).
>
> NIAF scales exceptionally well architecturally, as it predicts fixed-size modulation vectors for a continuous SIREN, keeping decoder's dimensionality constant regardless of chunk length. This decoupling enables upsampling to higher control frequencies (Appendix B.1) and offers a pathway to unify action chunks of varying physical durations across heterogeneous datasets (details in our reply to Reviewer jZrd).
>
> **Q3: Sensitive to SIREN architecture choices (depth and frequency)**
>
> While Appendix B.2 ablates chunk size, grouping, and activations, we agree depth and frequency parameters require clarification. For depth, we conducted new ablations on CALVIN ABC\_D, yielding average success lengths of 3.24 for depth 2, and 4.20 for depth 4. Our default of 3 hidden layers (4.47) provides sufficient representational capacity while preserving Florence-2's limited query token budget. For frequency, we use the standard SIREN default ($\omega_0=30$) as it stably control spectral sensitivity for high-frequency variations, and is a well-established default rather than the primary driver of our gains.
>
> **Q4: Higher-order derivatives necessary & B-splines.**
>
> In NIAF, position and velocity are the primary signals for impedance control. Higher-order derivatives mainly act as auxiliary smoothness regularizers.
>
> Guaranteeing $C^2$ or $C^3$ continuity in B-splines requires higher-degree basis functions, which widens their local support as each control point influences a larger trajectory segment. This results in difficulty in fitting localized, fine-grained motions. Conversely, NIAF's $C^\infty$ SIREN maintains arbitrary-order smoothness without losing high-frequency local details.
>
> We added a new smoothness-sensitive Water Cup Transport task to compare 4 Florence-2 policies under impedance control: NIAF (pos+vel supervision), alongside NIAF, BEAST, and FAST (all pos-only supervision). Velocity signals were obtained by derivatives for NIAF and BEAST, and via finite differencing for FAST. (Scenarios: [https://imgur.com/a/d81XpDq.png], Curves: [https://imgur.com/a/YF7OOLG.png]). Takeaways:
> * **Representation Superiority:** NIAF fits fine-grained motions better than B-spline based BEAST.
> * **Supervision Benefits:** Direct velocity supervision significantly improves execution smoothness over position supervision only methods.
>
> **Weakness 2: Clarification on CALVIN Improvements**
>
> We appreciate the opportunity to contextualize the CALVIN results. Since CALVIN's average chain length is an expectation over five consecutive tasks, this difference should not be interpreted as a simple per-task percentage improvement, as achieving additional gain demands reduction in compounding errors during long-horizon rollouts. Furthermore, CALVIN is highly saturated at the top end, improvements are difficult, especially for models without large-scale robot data pretraining or larger-scale backbones. That said, we will update the text to reframe the CALVIN performance as a robust validation of our method rather than a standalone claim.
>
> Thank you again for the constructive feedback. We believe these clarifications and new experiments will make the paper clearer and better calibrated.

---

> > ### Author Rebuttal · Reviewer_is7m · 2026-04-03
> >
> > I thank the reviewers for their rebuttal and the clarification on CALVIN saturation. I have two follow up questions:
> > - Figure 4: Could the authors clarify why OFT and BEAST cannot be evaluated on the latter tasks? In other words, please explain why they were "outside the scope of this portability test". Including all baselines on all tasks would provide a more complete and consistent comparison. As it stands, $\pi_{0.5}$+NIAF does worse than $\pi_{0.5}$ on Towel Folding (by 5%) and I would like to know how OFT and BEAST compare on this same task. I am not sure that porting NIAF onto one additional policy and evaluating it on two tasks -- one of which it performs worse than the base -- is sufficient to prove portability of NIAF.
> > - Follow-up on the Water Cup Transport experiment: Can the authors clarify how this experiment demonstrates why the higher order derivatives are actually necessary? The experiment only shows NIAF with pos or pos+vel supervision, but not, for example, acceleration or jerk supervision. This only demonstrate benefits from first-order (velocity) supervision but if higher order information is available implicitly in the INR, this experiment does not explicitly show where this information is actually useful. Further, can BEAST (or other baselines) be trained with velocity supervision, and if so, how does it perform under the same setting?

---

> > > ### Author Response · Authors · 2026-04-06
> > >
> > > We thank the reviewer for the thoughtful follow-up questions. We have conducted the requested experiments and provide clarifications below to address your concerns directly.
> > >
> > > **Q1. Evaluation of OFT/BEAST on the $\pi_{0.5}$ Backbone and the Portability of NIAF**
> > >
> > > We implemented OFT and BEAST on the $\pi_{0.5}$ backbone and evaluate them on Shape Insertion and Towel Folding. For $\pi_{0.5}$-BEAST, we replaced the flow-matching target with B-spline control points. For $\pi_{0.5}$-OFT, we projected each query to one timestep's joint action. The results are:
> > >
> > > **Table A: Success Rates**
> > >
> > > | Method | Shape Insertion | Towel Folding |
> > > | :--- | :---: | :---: |
> > > | $\pi_{0.5}$ | 75 | 90 |
> > > | $\pi_{0.5}$-NIAF | 82 | 85 |
> > > | $\pi_{0.5}$-BEAST | 40 | 85 |
> > > | $\pi_{0.5}$-OFT | 63 | 70 |
> > >
> > > **Analysis:**
> > > 1. Shape Insertion: the insertion phase contains critical micro-adjustments in demonstrations, as the clearance between the block and base is tight. $\pi_{0.5}$-NIAF performs best because it offers fitting capability over high insertion precision and local corrections near contact. Conversely, $\pi_{0.5}$-BEAST acts as a low-pass filter, it over-smooths these essential fine-grained motions and leads to insertion failures.
> > > 2. Towel Folding: success relies on long-horizon bimanual coordination and handling deformable objects, and all variants underperform base $\pi_{0.5}$ in this task. We attribute this to the fact that replacing the action head of a strong pretrained policy weakens its acquired knowledge priors and introduces some degree of forgetting. Moreover, $\pi_{0.5}$-BEAST and $\pi_{0.5}$-NIAF outperform $\pi_{0.5}$-OFT because their structurally continuous representations better maintain inter-step consistency, whereas OFT independently regresses each action step.
> > >
> > > **Addressing Portability:**
> > >
> > > The $\pi_{0.5}$ results are not our only evidence for portability. As detailed in Table 3 of our manuscript, we integrated NIAF into the Qwen3-VL backbone and compared it against FAST, OFT, Pi and GR00T, and NIAF achieved 3 highest results among four LIBERO suits. Thus, NIAF has bee integrated into encoder-decoder (Florence-2), decoder-only (Qwen3-VL), and VLM + action expert ($\pi_{0.5}$) architectures. We believe these results strongly supports NIAF's broad portability.
> > >
> > > **Q2. The Necessity of Higher-Order Derivatives and Velocity Supervision for BEAST**
> > >
> > > To clarify the role of higher-order terms, we added comparisons of NIAF with position and velocity supervision, alongside analytical jerk regularization. We also replaced BEAST's Cross-Entropy loss with L2 supervision on position and velocity, and include this comparison as well.
> > >
> > > We provide full-episode results at [https://i.imgur.com/HTbqzIU.png], and pairwise comparisons at:
> > > * BEAST (pos) vs. BEAST (pos+vel): [https://i.imgur.com/L3QfIso.png]
> > > * NIAF (pos) vs. NIAF(pos+vel) vs. NIAF(pos+vel+jerk): [https://i.imgur.com/euoQmhb.png]
> > > * NIAF(pos+vel) vs. BEAST (pos+vel): [https://i.imgur.com/mizlJtw.png]
> > >
> > > **Analysis:**
> > > 1. Velocity supervision effectively mitigating overfitting to demonstration jitter, reduceing peak velocity spikes in both policies.
> > > 2. Jerk regularization enables NIAF (pos+vel+jerk) to produce smoother position and velocity curves.
> > > 3. NIAF (pos+vel) yields a smoother whole-episode trajectory and fewer velocity spikes than BEAST (pos+vel), due to its superior action reconstruction capability.
> > >
> > > Since the above full-episode curves are somewhat cluttered, we conducted a more controlled toy fitting experiment. We simultaneously fit position and velocity for a single action chunk of length 200, using a SIREN network (same as in NIAF: 3 hidden layers, width 64) and a 4th-order B-spline (with control points doubled from 10 to 20 to strengthen its fitting capacity). We provide fitting results at:
> > > * SIREN (pos+vel) vs. B-spline (pos+vel): [https://i.imgur.com/cQeuPQ4.png]
> > > * SIREN (pos+vel+jerk) vs. B-spline (pos+vel): [https://i.imgur.com/hYqtEVM.png]
> > >
> > > **Analysis:**
> > >
> > > SIREN accurately reconstructs fine-grained local corrections, and adding jerk regularization smooths velocity profile while retaining positional accuracy. Conversely, **the B-spline achieves coarse smoothness at the expense of local precision**. It underfits micro-adjustments even with doubled capacity, directly explaining its performance collapse in precision-demanding tasks like Shape Insertion.
> > >
> > > Ultimately, velocity supervision ensures stable control, while jerk supervision guarantees further smoothness. NIAF's core architectural advantage over B-splines is its ability to decouple smoothness from expressiveness: **it delivers high-fidelity local precision while simultaneously enabling analytical smoothness regularization**. This combination offers smaller long-horizon compounding errors, precise micro-adjustment execution, and smoother impedance control.
> > >
> > > Thank you again engaging deeply with our work. We believe these additional experiments make the empirical picture clearer.

---

### Official Review · Reviewer_jZrd · 2026-03-13

**Soundness:** 3
**Presentation:** 2
**Significance:** 3
**Originality:** 3
**Overall Recommendation:** 4
**Confidence:** 2

**Summary:**

This paper proposes Neural Implicit Action Fields (NIAF), a continuous-time action representation for VLA-style policies. Instead of predicting discrete action chunks or compressed waypoint sequences, the policy predicts modulation parameters for an implicit SIREN-based action function, which can then be queried at arbitrary time resolutions and analytically differentiated to obtain velocity, acceleration, and jerk. The paper evaluates the approach on CALVIN, LIBERO, and real-world manipulation tasks, and argues that this representation is better aligned with physically continuous execution and impedance-style control.  The core idea—moving from discrete action sequences to continuous-time action functions—is meaningful and potentially useful. My main reservation is that some of the framing feels a bit stronger than what is fully established, and I would encourage the authors to more clearly separate the central representation idea from the specific implementation choices.

**Compliance With Llm Reviewing Policy:**

Affirmed.

**Final Justification:**

The rebuttal and follow-up responses addressed my main concerns satisfactorily. I especially appreciate the clearer separation between the core idea of continuous functional action representation and the specific design choices used in this paper, as well as the added clarification regarding comparisons to alternative continuous parameterizations. These responses improved my confidence in the paper’s main contribution.

I still think the paper is strongest as a representation-level reformulation, and that some of the broader framing should remain slightly more restrained. While the added evidence is helpful, the empirical picture is still not fully comprehensive regarding when this formulation is broadly preferable to simpler alternatives. However, I do not view these remaining issues as blocking.

Overall, I find the paper technically solid, meaningful, and sufficiently novel to merit acceptance, so I maintain my score of 4.

**Key Questions For Authors:**

First, while I like the continuous-time action function idea, I would appreciate a clearer discussion of alternative continuous action parameterizations. For example, how should readers think about NIAF relative to other plausible continuous representations such as splines, Fourier-style bases, or other implicit neural function classes? This would help clarify what is specific to NIAF versus what is more generally due to moving away from discrete waypoint prediction.

Second, the paper suggests that continuous functional action representations may have implications beyond decoding, but it does not really explore them. I would be interested in a brief discussion of how this representation might be used for planning, for example through time reparameterization, smoothness-aware trajectory refinement, or as an interface between high-level planning and low-level execution. I do not think this needs to be an additional experiment, but some discussion would strengthen the broader picture.

**Limitations:**

Yes.

**Strengths And Weaknesses:**

Strengths:
- The paper studies a meaningful and relatively underexplored issue: the mismatch between discrete action representations and physically continuous execution. That is a worthwhile problem to highlight.
- The method is conceptually clean. Representing an action chunk as a continuous function, rather than a fixed discrete sequence, is a genuine shift in output representation.
- The paper does more than just report benchmark gains. It also connects the representation to analytic derivatives and execution-side considerations, which gives the work a stronger systems flavor than many purely decoder-level changes.
- The empirical results are strong across both simulation and real-robot settings.

Weakness:

The main idea is promising, but some of the presentation feels slightly overpackaged. In my view, the strongest contribution is the continuous functional action representation itself; the broader “paradigm shift” framing is more ambitious than what is fully demonstrated.
Relatedly, the current method bundles together several ingredients: continuous-time action functions, SIREN as the function class, grouped hyper-modulation, and, in real-world experiments, higher-order derivative supervision.
Because these are introduced together, it is still somewhat unclear which parts are most essential to the gains.

---

> ### Author Rebuttal · Authors · 2026-03-31
>
> **Reply to Reviewer jZrd**
>
> Thank you for the thoughtful and constructive feedback. We are encouraged that you find the core idea, representing an action chunk as a continuous time function, as meaningful and potentially useful. In the revision, we will tone down the "paradigm shift" with "reformulate action representation", and restructure the Introduction and Method to make explicit that:
> 1. the continuous functional representation is the core claim,
> 2. SIREN with grouped hyper-modulation is our chosen instantiation,
> 3. derivative supervision is an extension for real-robot deployment enabled by this reformulation.
>
> The ablation in Table 6 (ReLU vs. Sine) and Figure 3 (NIAF vs. BEAST-CT) already partially isolate these layers empirically.
>
> **Q1: How should readers think about NIAF relative to other continuous action parameterizations, such as B-splines, Fourier bases, or other implicit neural function classes?**
>
> We agree this should be clarified better. The key distinction lies at the representation level, not in the choice of SIREN itself. A SIREN (more broadly INRs) modulated by hypernetworks provides a learned, input-adaptive function family, rather than relying on a basis fixed in advance as in B-splines and Fourier bases. Concretely, the shape of the action function is determined jointly by shared meta-parameters (a motion prior) and context-dependent modulation vectors predicted from the observations and instructions; thus the function family itself is conditioned on the task context. This is the structural difference we consider most important, and it gives NIAF stronger trajectory fitting flexibility within the action chunk space.
>
> We chose SIREN because it yields a smooth function that is analytically differentiable to arbitrary order, enabling unified position, velocity and acceleration supervision or regularization. We do not want to overstate this as a unique claim for SIREN, as other continuous parameterizations may also support this in principle. Our claim is that SIREN provides a particularly convenient and unified way to do this within the same end-to-end model, and our real-world results suggest that this design is practically useful.
>
> Empirically (Fig. 4), NIAF outperforms continuous action baselines like BEAST-CT and OFT on CALVIN ABC\_D. This suggests gains stem not only from avoiding tokenization or quantification but also from the learned implicit function's expressivity. A recent preliminary study also evidences that INRs can fit sparsely sampled continuous signals more precisely than cubic B-splines under matched regularization, although we cite this only as supporting intuition rather than definitive evidence in robot action learning (Yu et al., 2026). For further discussion on this, please refer to our reply to reviewer is7m (Q4).
>
> **Q2: Beyond decoding, what potential does this continuous representation have for planning, for example via time reparameterization or smooth trajectory refinement?**
>
> We appreciate you highlighting this. While our primary focus is on decoding and execution sides, continuous action representations offer several potentials as a planning interface.
>
> What we have in mind is that it supports time reparameterization naturally. Since the policy predicts a continuous function $A(\tau)$, a planner or execution module can dynamically stretch or compress the query schedule over $\tau$ to alter execution speed near contact areas without changing the underlying prediction. Additionally, whereas refining discrete waypoints requires editing a sampled sequence and then interpolating, a differentiable function representation allows constraints like smoothness or collision avoidance to be formulated and optimized directly in the function space.
>
> From a broader perspective, this continuous representation could be particularly valuable for unifying heterogeneous pretraining data. In large-scale pretraining where datasets are collected at varying frame rates or control frequencies, fixed-length discrete action chunks correspond to different physical durations. With NIAF, chunk supervision can instead be unified in physical time, for example representing one second of motion, ensuring learned motion patterns are temporally aligned across all datasets.
>
> Finally, we also clarify that by "bridging high-level prediction and low-level control," we simply mean mapping high-level multimodal context directly to a low-level executable continuous action function, rather than claiming improved VLM reasoning, long-horizon planning, or task decomposition abilities.
>
> Thank you again for the constructive feedback, which will undoubtedly make the paper clearer and better calibrated.
>
> **References:**
>
> [1] Yu, H., et al. Comparing Implicit Neural Representations and B-Splines for Continuous Function Fitting from Sparse Samples. arXiv preprint arXiv:2602.20535, 2026

---

> > ### Author Rebuttal · Reviewer_jZrd · 2026-04-02
> >
> > The rebuttal addressed my main concerns satisfactorily. In particular, I appreciate the clearer separation between the core representation idea and the specific implementation choices, as well as the effort to better calibrate the paper’s framing. I still think that broader comparisons across alternative continuous action parameterizations would further strengthen the work, but I do not see this as a blocking issue. My overall recommendation remains 4.

---

> > > ### Author Response · Authors · 2026-04-06
> > >
> > > We sincerely thank the reviewer for the continued support and for confirming that our rebuttal addressed your main concerns.
> > >
> > > Regarding your note on broader comparisons across alternative continuous action parameterizations, we completely agree on its importance. We would like to gently note that our baseline BEAST fundamentally represents actions using B-splines, which serves as our primary comparison against an alternative continuous parameterization.
> > >
> > > To explore this comparison deeper, we have conducted a new set of controlled experiments in our latest follow-up reply to **Reviewer is7m (Q2)**. We explicitly compare our SIREN-based representation against a high-capacity B-spline for simultaneously fitting position and velocity. As detailed in that reply, the new visual and empirical results demonstrate that **the B-spline achieves coarse smoothness at the expense of local precision**, it acts as a low-pass filter that underfits critical micro-adjustments (which explains its performance drop on precision-demanding tasks like Shape Insertion). In contrast, the architectural advantage of NIAF is its ability to decouple smoothness from expressiveness: it provides high-fidelity local expressiveness while simultaneously allowing for analytical smoothness regularization.
> > >
> > > We will add these comparisons and visual demonstrations in the revision to thoroughly address this aspect. Thank you again for your time and constructive guidance, which has greatly strengthened our work!

---

### Decision · Program_Chairs · 2026-04-30

**Decision:**

Accept (regular)

**Comment:**

This paper proposes Neural Implicit Action Fields, a continuous-time action representation for VLA policies.  All four reviewers recognize that addressing the mismatch between discrete action sequences and the core idea of predicting a continuous action function is conceptually novel. The authors addressed most concerns during rebuttal, clarifying the separation between the core representation and specific design choices. The final sores are three Weak Accepts, one Weak Reject (the only negative review mention “I am providing a final rating of weak accept”). Thus, the Area Chair recommends acceptance as Accept.